# ALIGNING AGENTS LIKE LARGE LANGUAGE MODELS

## ABSTRACT

Training agents to behave as desired in complex 3D environments from visual information is challenging. Imitation learning from diverse human behaviour provides a scalable mechanism for training an agent with generally sensible behaviours, but such an agent may not perform the specific behaviours of interest when deployed. To address this issue, we draw an analogy between the undesirable behaviours of imitation learning agents and the unhelpful responses of unaligned large language models (LLMs). We then investigate how the procedure for aligning LLMs can be applied to aligning agents from pixels in a complex 3D environment. For our analysis, we utilise an academically illustrative part of a modern console game in which the human behaviour distribution is diverse, but we would like our agent to imitate a single mode of this behaviour. We find that we can align our base agent to consistently perform the desired behaviour, providing a demonstration of a general approach for training agents to perform specific behaviours in complex environments.

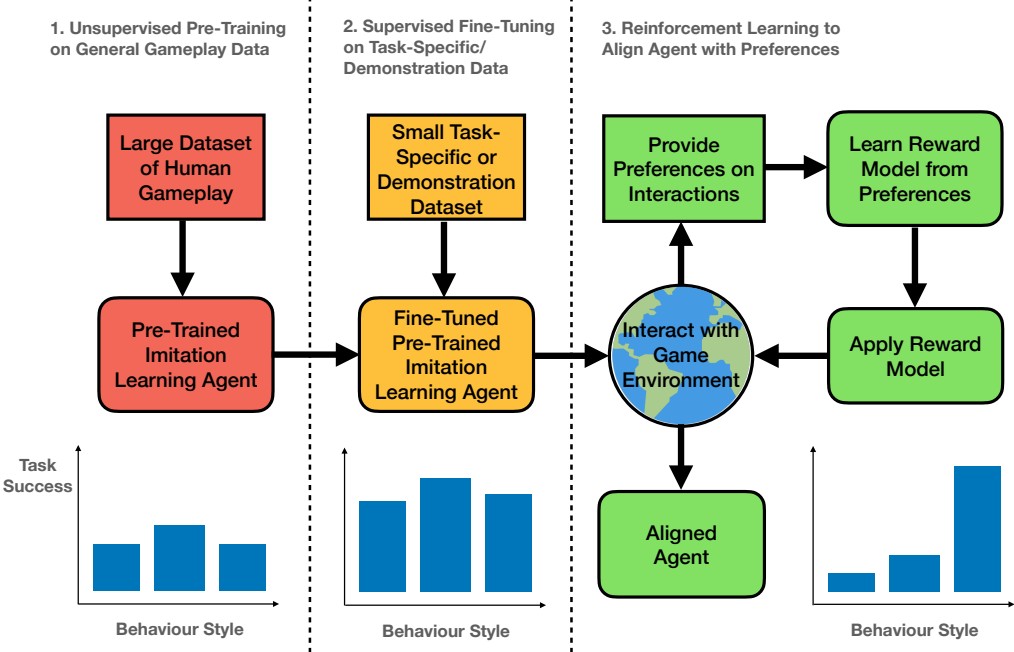

Figure 1: Illustration of our approach for aligning generally capable agents with a game designer's tasks, goals or preferences. A general agent pre-trained to imitate a large diverse dataset of human gameplay provides a base agent which can be more robustly fine-tuned to imitate a smaller task or demonstration dataset. This agent can then be further fine-tuned with reinforcement learning using a reward model learned from preferences to reliably achieve a behaviour with the desired style. This approach is analogous to the alignment procedure used for recent large language models.

# 1 INTRODUCTION

The optimal recipe for training generally competent agents to act in complex 3D environments from visual information is an open question. Many modern console games provide such 3D environments, where the state space, action space and temporal horizons are large enough that learning from scratch is usually infeasible, even with a clearly defined goal. In these environments, a natural approach is to leverage a large dataset of general human behaviour to pre-train an agent with imitation learning. This provides an agent with a general understanding of player behaviour, and there is evidence of generalisation benefits from scaling up data and compute (Reed et al., 2022; Baker et al., 2022).

However, such an agent will inevitably learn to imitate all behaviours found within the human game-play, including undesirable behaviours of novice or malicious players, analogous to unhelpful or toxic responses of unaligned large language models (Ziegler et al., 2020). Additionally, game designers may have preferences for agents to act with a certain style or strategy, for which it may be difficult to codify a suitable reward function (Aytemiz et al., 2021). There is a parallel between training useful agents and aligning large language models. In the same way that LLMs have been aligned to serve as customer service chatbots (Banerjee et al., 2023), search engines (Spatharioti et al., 2023) and code generation assistants (Chen et al., 2021b), we can imagine aligning a large imitation learning agent with various objectives within a game. For example, we may wish to train agents that can act as helpful allies, appropriately challenging opponents, or non-player-characters (NPCs) with various objectives.

In this work, we take inspiration from recent success with the current procedure for aligning large language models (LLMs) and take a first step towards investigating the key considerations for applying this procedure to align imitation learning agents. To address this question, we focus our analysis on an academically illustrative part of a modern console game, where the human behaviour distribution is distinctly multi-modal over non-negligible time horizons (of the order of $\sim 10$ seconds), but we desire our agent to imitate a single mode of this distribution. This behaviour would be difficult to learn from scratch, but an imitation learning agent pre-trained on general behaviour data only sometimes performs the desired behaviour.

Specifically, we focus on a part of the game where players must travel from multiple spawn points to one of three jumppads which launch the player onto different parts of a central island. We find that a base imitation learning agent learns to model the full human distribution and reaches all three jumppads in similar proportions to the human data. We then demonstrate that we can align this agent to consistently reach a single preferred jumppad, using synthetic preference labelling to train a reward model and online reinforcement learning using the reward model. We perform this entire procedure from visual inputs with full controller action output, maintaining the generality of our approach to other games, and providing a demonstration of applying the modern large language model training paradigm to aligning agents on a real 3D game.

# 2 RELATED WORK

## 2.1 LARGE SCALE IMITATION LEARNING AGENTS

Imitation learning and offline reinforcement learning (Levine et al., 2020) have been gaining popularity in recent years with the aim of demonstrating similar scaling laws to LLMs (Kaplan et al., 2020; Hernandez et al., 2021) to provide a path to obtaining more generalisable agents on ever more complex environments. Decision Transformer (Chen et al., 2021a) proposed learning a transformer-based policy from offline data that can be conditioned on a desired return. Multi-game Decison Transformers (Lee et al., 2022) extended this approach to learn multiple game policies with a single model, providing evidence that scaling transformer policies with diverse data leads to performance improvements. GATO (Reed et al., 2022) demonstrated that the tasks do not need to be limited to games, and was able to complete most of its 604 diverse training tasks to a rudimentary level. RT1 (Brohan et al., 2023b) shows similar scaling potential for transformers on robotics tasks, while RT2 (Brohan et al., 2023a) integrated vision-language models to help with zero-shot generalisation to new tasks. In addition to transformer policies, Kumar et al. (2022) show promising scaling for offline Q-learning, and Pearce et al. (2023) demonstrate that diffusion models can be utilised to better capture complex human action distributions. While these methods demonstrate the scaling potential of imitation learning, they do not incorporate preferences for imitation of multi-modal behaviours.

## 2.2 Fine-tuning Imitation Learning Agents with Reinforcement Learning

Fine-tuning large imitation learning models has also demonstrated impressive results in recent years. The use of imitation learning as pre-training for reinforcement learning was first investigated to improve the sample efficiency of deep RL algorithms by reducing the exploration space (Hester et al., 2017; Vecerik et al., 2018). AlphaGo (Silver et al., 2016) and subsequently AlphaStar (Vinyals et al., 2019) demonstrated that scaling up imitation learning on human data could provide a strong behaviour prior for performing reinforcement learning fine-tuning (to maximise the win-rate) in environments like StarCraft where reinforcement learning from scratch is infeasible. VPT (Baker et al., 2022) extended this paradigm to web-scale data by first training an inverse dynamics model on MineCraft videos and then performing imitation learning on 70k hours of human gameplay. By fine-tuning with reinforcement learning on task-specific rewards, the agent was the first to be able to craft diamond tools. While these works demonstrate the potential of fine-tuning large imitation models, they use hard-coded reward functions to maximise agent performance rather than align an agent's behaviour with subjective preferences.

## 2.3 Reinforcement Learning from Human Feedback for Agents

Training agents with human preferences has a long history, as reviewed by Wirth et al. (2017) and Zhang et al. (2021) (including notably Bennett et al. (2009) and Bradley Knox & Stone (2008)), leading to the popular modern formulation for deep learning proposed by Christiano et al. (2017). This formulation involves training a reward model from human preferences using a Bradley-Terry model (Bradley & Terry, 1952) to scale up costly human feedback, and then using that reward model to train an agent with reinforcement learning to align its behaviour with the human preferences. This idea was extended by Ibarz et al. (2018) to include imitation learning as pre-training to improve the efficiency of early preference learning. PEBBLE (Lee et al., 2021) instead utilises unsupervised pre-training rather than imitation learning to increase the diversity of initial behaviours for preference labelling. More recently, Abramson et al. (2022) demonstrated the scalability of reinforcement learning from human feedback at improving the task-completion success of imitation agents where tasks are specified by natural language in a 3D simulated world. Rather than train a single capable agent, in this work we envision a generally capable base agent that can be aligned by game designers or end-users to perform different tasks or behave with different styles according to potentially intangible preferences (Aytemiz et al., 2021; Devlin et al., 2021).

## 2.4 Reinforcement Learning from Human Feedback for LLMs

The capabilities of large language models have also developed significantly in recent years, with overlap with decision-making agents that has inspired developments on both fronts. Radford et al. (2018) demonstrated that pre-training a language model with an unsupervised generative task (predicting the next token) on a large diverse corpus of text, followed by fine-tuning on a specific task dataset led to significantly improved performance compared to training on the task-specific dataset alone. Stiennon et al. (2020) (following on from Ziegler et al. (2020)) then popularised the use of reinforcement learning from human feedback (RLHF) to further fine-tune these responses, by demonstrating that models aligned with human preferences could better summarise long text passages. This procedure was then applied to train language models to follow instructions, leading to InstructGPT (Ouyang et al., 2022), Chat-GPT (OpenAI, 2022) and subsequently GPT-4 (OpenAI, 2023) models. This procedure has also been successful in open-source reproductions of these models, such as Open Assistant (Köpf et al., 2023), Claude (Bai et al., 2022) and Llama 2 (Touvron et al., 2023). This recipe has also demonstrated success in generative vision tasks, where it has been shown that generative pre-training has similar benefits for images (Chen et al., 2020), and recently that RLHF can similarly be utilised to better refine state-of-the-art image generation models (Xu et al., 2023; Black et al., 2023). While the components of this procedure have been applied to behaviour models (i.e. agents) both individually (Christiano et al., 2017; Reed et al., 2022) and in combination (Ibarz et al., 2018; Abramson et al., 2022), in our work we aim to investigate the benefits of applying the entire modern LLM alignment procedure to agents trained end to end from visual inputs to a general action space to generate multiple preference aligned agents from a large base model.

## 3 GAME ENVIRONMENT AND ALIGNMENT GOAL

For our analysis, we utilise the video game Bleeding Edge[1], which was launched in 2020 for Xbox One[2]. It is a team-based 4v4 online multi-player video game. Players select from thirteen possible heroes, each with different abilities. The game is played with a third-person view, with the camera angle controlled by the player, so the environment is partially observable. Here we focus on a single map, called Skygarden. This map is spread over three islands each with multiple elevation levels, including a main island and two launch islands (one for each team).

### 3.1 ALIGNMENT OBJECTIVE

At the beginning of the game, players spawn at one of four nearby points on their team's launch island. As players compete on the main island, they may be attacked by opposing team players, also leading to them respawning at one of these four points. From the spawn location the player can navigate across the launch island to one of three jumppads that launches the player from their launch island onto the main island. Depending on the jumppad selected, the player will be launched onto different areas of the main island, so players may wish to take different jumppads during a match depending on the location of opposing team players. For the purpose of this work, we aim to train an agent to consistently navigate across the launch island to a single one of the three jumppads.

This small part of the full game therefore provides an academically illustrative task in which the human data distribution is distinctly multi-modal, but we would like our agent to learn a single mode of behaviour. Since the task requires around 60 consistent actions to complete for an optimal agent, a random agent will rarely leave the spawn area. Additionally, without access to privileged information such as the agent location (we only provide visual input as described below) and an externally shaped reward function, it would be difficult to train an agent to complete this task. On the other hand, an imitation learning agent with a suitable objective will learn to reach all three of the jumppads, as in the human distribution. Therefore while the task is simple, it provides a clear motivation for alignment and a setting which we can concretely analyse.

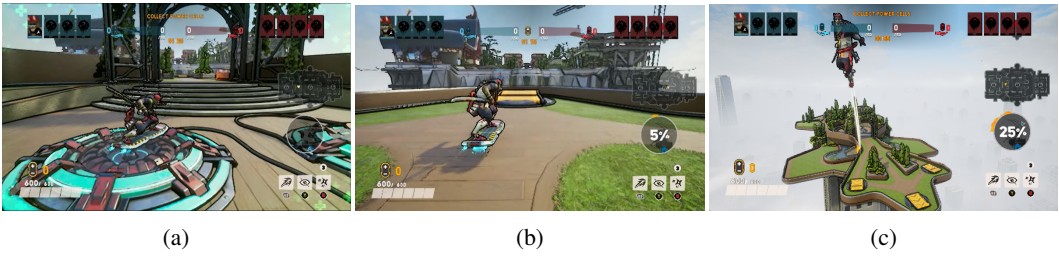

|       (a)       |       (b)       |       (c)       |

Figure 2: Screenshots of the agent at a spawn point (2a), heading towards the middle jumppad (2b), and an agent looking back at the launch island after having launched from the right jumppad (2c)

### 3.2 OBSERVATION AND ACTION SPACES

To maintain the generality of our approach to any 3D environment, we take visual gameplay observations $\mathbf{o}_t \in \mathbb{R}^{H \times W \times 3}$ as input, sampled at 10Hz. We do not provide any privileged information, so the agent only has access to input information available to human players. The action space consists of 12 action commands and two joysticks. The left joystick controls the movement of the agent, while the right joystick controls the camera angle. We decompose each joystick into an $x$ and a $y$ component which are independently discretised into 11 buckets. This creates a $12 + 2 + 2 = 16$ dimensional discrete action space in total, although the joystick dimensions are of most relevance to our objective.

---

[1]https://www.bleedingedge.com/en
[2]https://www.pcgamingwiki.com/wiki/Bleeding_Edge

## 4 IMPLEMENTATION AND ANALYSIS

We now follow the general procedure outlined in Figure 1 and describe our specific implementation of this procedure for obtaining an agent aligned to reliably head towards and reach a preferred jumppad. We analyse the key components of this procedure to provide insight into the practicalities of aligning agents with preferences in a 3D environment from visual input.

### 4.1 TRAINING A BASE IMITATION LEARNING MODEL

We aim to use a minimal but scalable approach for training a base imitation learning model from visual observations to predict gamepad actions. This provides a behavioural prior with a general understanding of human gameplay. Specifically we train a transformer autoregressively to learn a policy $p(a_t|o_t, ..., o_{t-H})$ using a cross-entropy loss. For the purpose of this work, we consider an agent trained on diverse data within a particular game, but note that given our unified observation and action spaces (such as the visual observations and gamepad actions we consider), it would also be possible to train across games, as explored in previous work (Reed et al., 2022; Lee et al., 2022).

**Dataset:** For general pre-training, a dataset was extracted from recorded human gameplay, as explained in Appendix C. This large unfiltered dataset consists of 71,940 individual player trajectories from 8788 matches recorded between 09-02-2020 and 10-19-2022, which amounts to 9875 hours (1.12 years) of individual gameplay. For the purposes of this work, we utilise an agent that was trained for less than one epoch on this dataset.

**Architecture and Training:** For the policy, we use a GPT-2 (Radford et al., 2019) causal transformer architecture with 103M parameters, similar to that used by VPT (Baker et al., 2022). Observations from the human gameplay $\mathbf{o}_t \in \mathbb{R}^{H \times W \times 3}$ are taken directly as input to a convolutional encoder to give observation embeddings $\mathbf{z}_t$. The transformer is trained with a context window of $H = 32$ timesteps (corresponding to around 3s of gameplay given the 10Hz sampling). The context window is important since the game is partially observable: as the context window is increased, the agent is able to better capture the state of the environment and take more informed actions (i.e. with a more Markovian state) at the cost of computational complexity. The output corresponds to the 16 discrete action dimensions. The transformer and convolutional encoder are both trained end to end with a cross-entropy loss over all output action components to provide a policy $p(a_t|o_t, ...o_{t-H})$.

$$\mathcal{L}(\pi) = -\sum_{\tau \in D} \sum_t \log \pi(a_t^\tau | o_t^\tau, o_{t-1}^\tau, ...) \tag{1}$$

**Evaluation:** Once trained, we ran our base pre-trained agent online in the game environment and recorded which jumppad was reached for 1000 episodes. To run the agent online, we initialised the Ninja character with an empty context buffer at one of the four spawn points on a single launch island at random. We ran the agent until it reached one of the jumppads or timed out after 100 timesteps. The jumppad distribution reached is demonstrated in Figure 3 below.

We find that our base model reaches a jumppad $56.4\%$ of the time, and has a bias towards the middle jumppad. Our base model therefore provides a reasonable behavioural prior that is much better than random exploration of the launch island (which has a $0\%$ success rate). However, the success rate could be improved, which is likely partly due to under-training of our base model, and partly due to distribution shift between the offline data and the online environment. For example, the base model was trained on data containing all thirteen possible characters, while our online evaluation only

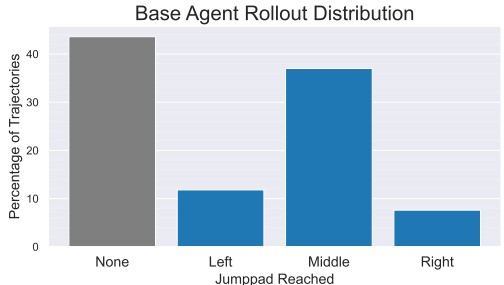

Figure 3: Distribution of jumppads reached by the base imitation learning agent.

uses the default Ninja character. Additionally, we initialise our agents online at the spawn point with an empty context buffer, while in training the agent will have access to context including a spawn animation and prior gameplay. While measures can be taken to avoid this offline to online shift, some distribution shift is usually unavoidable as we discuss further in Appendix E.

## 4.2 SUPERVISED FINE-TUNING ON TASK RELEVANT DATASET

Following the LLM pipeline, we now fine-tune our agent on a curated dataset of the task we wish to imitate. Pre-trained transformer models have been shown to fine-tune more effectively, essentially increasing the size of the fine-tuning data compared to training from scratch (Hernandez et al., 2021). For this work we focus on fine-tuning on a task relevant subset of the pre-training data. However, we note that for specific behaviours or tasks, this could also consist of fine-tuning on demonstrations of the desired behaviour by the game designer.

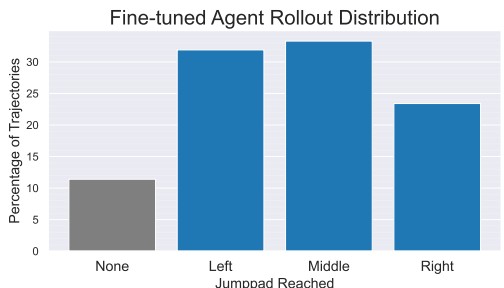

Figure 4: Distribution of jumppads reached by the fine-tuned imitation learning agent.

**Dataset and Training:** To obtain a task relevant dataset, we curated 300 trajectories (100 successful trajectories per jumppad) for fine-tuning, all of which involved the character Ninja and were filtered to contain only the part of the trajectory on the launch island after the first respawn until a jumppad was reached.

**Results:** We evaluated our agent in the environment again for 1000 episodes, following the same procedure as before. We see in Figure 4 that the success rate of reaching the jumppad is now significantly higher ($88.6\%$), and that the agent reaches all three jumppads around a third of the time (corresponding to the dataset we fine-tuned on). As an ablation, we also considered training the model from scratch on the fine-tuning dataset, but found that agent performance is less robust than fine-tuning the pre-trained model, as shown in Appendix D.4.

## 4.3 OBTAINING PREFERENCE DATA ON ONLINE ROLLOUTS

Once the policy has been fine-tuned on the behaviours of interest, it can be rolled out online multiple times to generate trajectories for human feedback. We use the fine-tuned agent for these since it provides more suitable trajectories for soliciting preferences compared to the base agent. This is analogous to generating multiple LLM responses to a prompt, but in this context the prompt becomes the initial observation (and optionally context of previous observations and actions provided). Similarly to LLMs, there are tradeoffs to be made in the diversity, quality and quantity of preferences when provided by humans (discussed further in Appendix J), but here we avoid such issues by utilising synthetic preferences to isolate the effect of quantity.

**Online Rollouts:** Equivalent to the previous evaluation, we initialised the Ninja character with an empty context buffer at one of the four spawn points on a single launch island at random, and ran the agent until it reached one of the jumppads or timed out after 100 timesteps. We repeated this procedure to generate a video dataset of 2400 on-policy trajectories to be used for preference labelling and analysis. We subsequently divided these trajectories into train and test datasets, with 1000 trajectories for training and 1400 for evaluation.

**Generating Preferences:** For the purposes of our analysis, we utilise synthetic preferences based on the primary criteria:

*Preferred Jumppad Reached > Other Jumppad Reached > No Jumppad Reached*

Within each of these primary categories, we further rank trajectories by their duration, with shorter trajectories being preferred. By selecting variable numbers of trajectories within the training dataset, we are then able to investigate how reward model performance scales with number of comparisons (which is a proxy for the human labelling time requirement). The preference dataset is then created by considering all pairwise comparisons utilising the criteria above for determining if one trajectory is preferred over another or if they are equal.

## 4.4 TRAINING A REWARD MODEL ON PREFERENCES

A reward model is then trained on these online trajectories such that the reward model provides higher reward for preferred trajectories. While this reward model is usually trained from scratch in

the context of agents, following the modern LLM procedure it is also possible to utilise the fine-tuned policy model by replacing the action classification head with a scalar regression head such that the reward model can also benefit from the pre-training, and share the same knowledge as the agent to reduce hallucinations (i.e. reduce out-of-distribution behaviours) (Ziegler et al., 2020).

**Dataset:** We use the video recordings of the online agent trajectories from spawn until jumppad (or timeout) together with pairwise comparisons obtained from the preferences $P$, $(\tau_A \succ \tau_B) \in P$.

**Procedure:** We follow the standard Bradley-Terry (Bradley & Terry, 1952) model procedure to train a reward model $\hat{r}$ from these pairwise preferences. Specifically, we interpret trajectory rewards as preference rankings analogous to Elo (Elo, 1978) rankings developed for chess, such that the annotator's probability of preferring a trajectory depends exponentially on the trajectory reward. We can then fit the reward model by minimising the cross-entropy loss between these probabilities and the preference labels (Christiano et al., 2017), which gives:

$$\mathcal{L}(\hat{r}) = \sum_{(\tau_w, \tau_l) \in D} -\log\left(\sigma\big(\hat{r}(\tau_w) - \hat{r}(\tau_l)\big)\right) \tag{2}$$

where $\sigma$ is the sigmoid function and $(\tau_w, \tau_l)$ are the trajectories being compared, with $\tau_w$ being the winning (preferred) trajectory and $\tau_l$ being the losing trajectory.

Since the reward model is only trained on comparisons, the scale of the predicted rewards is arbitrary. As a result, we found the need to apply a small amount of $L2$ regularisation to prevent the scale of the rewards becoming overly large for the best and worst trajectories considered. We further empirically normalise the reward model after training using the max and min of predicted rewards over the training trajectories to scale our reward model output to be in the range $\hat{r} \in [0, 1]$.

**Architecture:** Previous work on RLHF for agents has generally relied on simple (often linear) reward models (Bıyık et al., 2021). However recent work on LLMs has demonstrated that reward models that utilise the pre-trained or fine-tuned policy model with the action classification head replaced with a scalar regression head generally perform better (Stiennon et al., 2020), and also improve with scale (Ouyang et al., 2022; Touvron et al., 2023). Following this procedure, we extract embeddings from our agent to produce a $1024$ dimensional embedding for each timestep, which are fed into the reward model to produce a scalar return for the trajectory which is then input to the loss function in Equation 2. As an ablation, to investigate whether the representation of the observations used for the agent for imitation learning are also beneficial for learning the reward (i.e. predicting preferences), we consider a reward model with an equivalent architecture but with a randomly initialised encoder to provide random projections of the image as embeddings for each timestep. We further investigate how reward model performance scales with number of comparisons, to obtain an estimate of the human time required to provide sufficient feedback for training the reward model.

**Results:** To measure the performance of our reward models, we apply each reward model to our test trajectories. We compute the pairwise preferences according to the reward model, and compare to the ground truth pairwise preferences to obtain a test set preference accuracy. We provide our reward model performances in Figure 5.

We find that reward model accuracy generally increases with number of comparisons, although the random projection reward model has high variance. Importantly, the reward model utilising the agent encoder performs better than the reward model using random projections across the full range of comparison sizes, suggesting that the imitation learning agent's representations of the observations contains information beneficial to predicting preferences. We further find that when using this encoder it is possible to train a reward model from visual input to achieve over 90% preference accuracy with only $\sim 100$ comparisons. While we utilised synthetic preferences, we note that this would correspond to less than 1 hour of labelling time for our task.

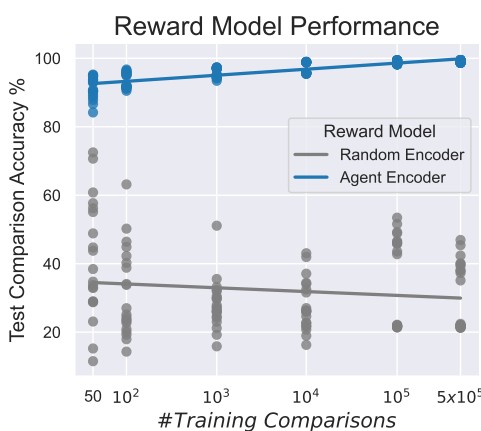

Figure 5: Reward model test performances.

### 4.5 Aligning the Fine-tuned Model with the Reward Model

Finally, we align our agent with our behaviour preferences as captured by our reward models.

**Procedure:** We run our fine-tuned agent in our environment as described in Section 3 to generate online rollouts. After each trajectory is complete, we apply our reward model to the trajectory to generate a reward corresponding to that trajectory. We then use this reward as the return for that trajectory and update our agent policy $\pi$ using a minimal undiscounted REINFORCE (Williams, 1992) loss. While the LLM literature commonly uses PPO (Schulman et al., 2017), we utilised REINFORCE for simplicity.

For our experiments we ran our agent online for 9600 episodes (corresponding to around 1 day of real time gameplay), using a batch size of 16 to give 600 online parameter updates, for 3 independent runs. We fine-tune only the last layer for our work, but note that modern approaches such as Low-Rank Adaption (LoRA) (Hu et al., 2021) used for fine-tuning LLMs could also be used here. We also investigated the use of an additional KL divergence term to regularise the optimised policy towards the initial fine-tuned policy, as is commonly used in LLM alignment (Touvron et al., 2023; Bai et al., 2022), but found it unnecessary in preliminary experiments.

#### 4.5.1 Aligning Agent Towards Left Jumppad

We begin by focusing on aligning our agent towards the left jumppad. We plot the average success rate of reaching the desired jump-pad against the # of episodes with reward models that have been trained on 100 up to 500k comparisons in Figure 6. We see that all of our reward models are sufficient to align our agent to consistently reach the left jumppad, with reward models trained on more data (that achieved higher test performances) generally leading to better alignment.

However, we see that the agents take most of the 600 updates to fully align in our simplistic setup, corresponding to 1-3 days of real time training. To improve this efficiency, we consider the addition of an additional preference fine-tuning phase on the preferred trajectories. Specifically, we apply the reward model to the trajectories it was trained on and take the top 20% of trajectories by reward (corresponding to the preferred trajectories) and perform additional fine-tuning on these trajectories. We note that this is equivalent to Reinforced Self-Training (ReST) concurrently introduced for aligning language models (Gulcehre et al., 2023) with a single iteration of filtered fine-tuning. We find that this improves online performance across updates and across

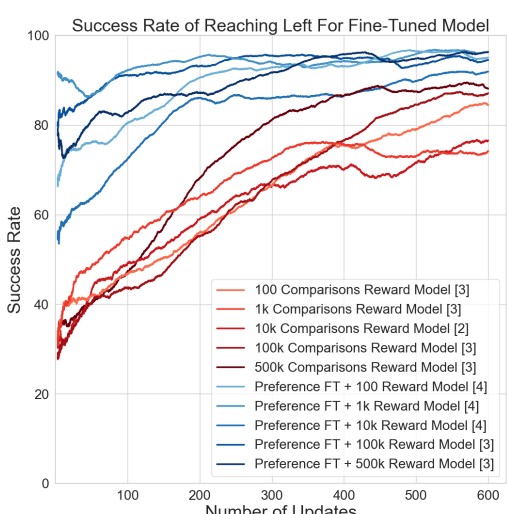

Figure 6: Left jumppad success rate during online alignment. Red lines show the fine-tuned agent from Section 4.2 aligned with reward models of varying performance (darker corresponds to more preference data). Blue lines correspond to additionally fine-tuning the agent on the 20% highest reward trajectories before online alignment. We see: 1) Higher accuracy reward models (using more preference data) generally lead to better alignment, 2) Preference fine-tuning improves performance across reward models and updates.

all reward models, shown in Figure 6 (fine-tuned agents shown in blue, with equivalent reward models to those shown in red). However, in the absence of additional iterations of this procedure, we find that subsequent reinforcement learning still improves alignment.

#### 4.5.2 Aligning Agent Towards Right Jumppad

We now consider aligning our agent towards the right jumppad. Since this task is seemingly of identical difficulty, we would expect to once again be able to easily align the agent. However, we can see in Figure 7 that both the fine-tuned and preference fine-tuned agents do not align as quickly towards the right jumppad, although they show similar trends in terms of initial preference

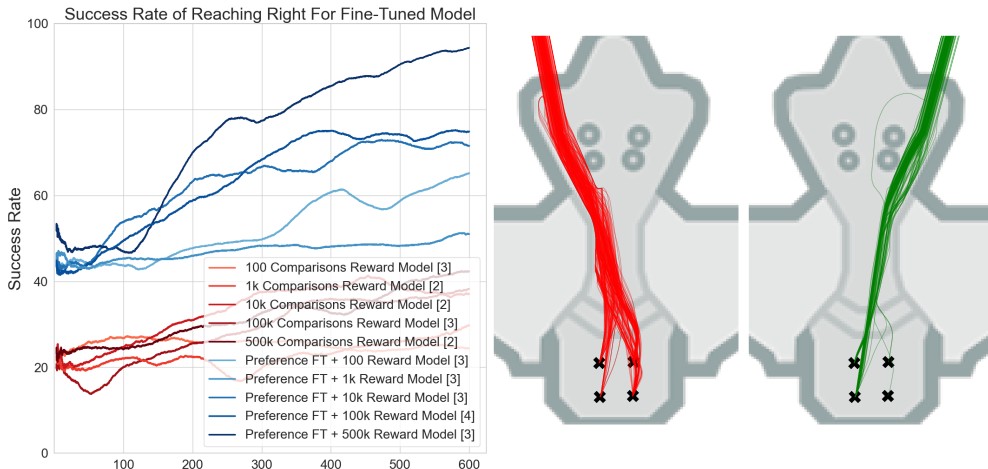

Figure 7: Right jumppad success rate during alignment, equivalent to Figure 6. Similar trends hold, but alignment is less effective.

Figure 8: Fine-tuned agent trajectories which reached the left (red) or right (green) jumppads. While fine-tuning enables more efficient preference labelling, it reduces diversity, potentially limiting alignment.

fine-tuning being beneficial for all reward models across alignment updates, and better performing reward models (using more preference data) generally aligning better.

In order to investigate the cause of this discrepancy, we analyse the rollouts of the fine-tuned agent in Figure 8. We see that for the trajectories that successfully reach the left jumppad, they start relatively evenly at all 4 possible spawn locations. However, of the trajectories that successfully reach the right jumppad, 84% originate from only 1 of the spawn locations (15% from another, and only 1% from the 2 spawns on the right hand side). Therefore it is unsurprising that it is more difficult for the agent to learn to go right, given that in half the episodes (corresponding to the right spawn locations) the agent rarely receives any positive signal (although more sophisticated RL algorithms may help to alleviate this difficulty). To confirm this hypothesis we demonstrate that the base agent can be aligned more symmetrically in Appendix G. Final jumppad distributions for our aligned agents are shown in Appendix H and a heatmap illustrating the alignment pipeline is shown in Appendix I.

These results highlight the importance of maintaining diversity of behaviour when attempting to align agents (Touvron et al., 2023; Casper et al., 2023). At every stage of the alignment pipeline, the agent becomes increasingly specialised and less diverse in its behaviour. While this generally leads to more efficient learning and alignment, it can also cause unintended outcomes such as the increased difficulty we find with aligning towards the right jumppad compared to the left. Furthermore, these results highlight the importance of the alignment procedure described, to gradually refine the behaviour at each stage. Additional research on improving the robustness and efficiency of this procedure for more unbalanced and limited datasets available in the context of agents will be important for practical application of the LLM paradigm to gameplaying agents and beyond.

## 5 CONCLUSION

In this paper, we have provided an analogy between training agents to perform desired behaviours in complex game environments and training large language models. We demonstrated a proof of concept for using this procedure to align agents with preferences from pixels on a modern console game. This enabled us to train an agent to achieve a specific behaviour in the game that would be difficult to achieve with either imitation learning, reinforcement learning or reinforcement learning from human feedback alone. Our analysis shows that many of the recent developments in the current procedure for training large language models, such as using a supervised fine-tuning step and using pre-trained reward models, can be applied and have similar benefits for training agents. We hope that our work encourages further communication and collaboration between the gaming and language model communities to enable shared insights and provide a path towards practical application of generally intelligent agents in modern games.

## ETHICS STATEMENT

This work considers aligning large scale imitation agents with human preferences. The preferences considered in this work are synthetic for analysis purposes, but in general the source of the preference data is an important ethical consideration to understand any biases in alignment. The data used for the base imitation learning agent is however real human data, for which data collection was covered by an End User License Agreement (EULA) to which players agreed when logging in to play the game for the first time. Our use of the recorded human gameplay data for this specific research was governed by a data sharing agreement with the game studio, and approved by our institution's IRB. To minimize risks regarding data privacy, any personally identifiable information (Xbox user ID) was removed when extracting the data used for this study from the original replays.

More generally, research into aligning agents with preferences is important to ensure that agents are helpful and harmless. However, this procedure has various known open problems and limitations (Casper et al., 2023). Games therefore provide an important test bed for such research, helping to mitigate risks and provide insights that may generalise to other applications.

## REPRODUCIBILITY STATEMENT

The general procedure proposed in this work (outlined in Figure 1 and discussed in Appendix A) is widely reproducible. Efforts have been made to ensure that all important implementation details for our application of this procedure are reflected in the paper (Section 4) and the authors are happy to disclose any details that may have been unintentionally missed. While we are unable to release the human gameplay data and environment for privacy and intellectual property reasons, we hope that our results and analysis provide insight into the potential benefits and challenges of applying this approach to agents in such environments. We also aim to inspire further research into each of the components of the pipeline, and reproduction of this work in other environments and domains.

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

## A  DISCUSSION OF GENERAL PROCEDURE FOR ALIGNING AGENTS

We break down our procedure for aligning agents with preferences (see Figure 1) into five steps:

1. **Train a Base Imitation Learning Policy**
   The first ingredient for training large language models is to train a large, scalable transformer architecture with self-supervised next-token prediction on a diverse dataset of human text to obtain a general understanding of the structure of human language and learn a language prior. In the context of agents on modern console games, we interpret this as imitation learning to predict the next action taken in human gameplay data to obtain a general understanding of human gameplay and learn a behavioural prior. Specifically this involves training a transformer autoregressively to learn a policy $p(a_t|o_t, ..., o_{t-H})$ using a cross-entropy loss. For the purpose of this work, we consider an agent trained on diverse data within a particular game, but note that given our unified observation and action spaces (such as the visual observations and gamepad actions we consider), it would also be possible to train across games, as explored in previous work (Reed et al., 2022; Lee et al., 2022).

2. **Supervised Fine-Tune on a Task Relevant Dataset**
   The next step in the current LLM pipeline is to fine-tune the foundation model on task relevant data, such as instruction data (Chung et al., 2022). Pre-trained transformer models have been shown to fine-tune more effectively, essentially increasing the size of the fine-tuning data compared to training from scratch (Hernandez et al., 2021). For decision-making agents this involves fine-tuning by imitation learning on the objective (or game) of interest, although may not be required if the base agent is already behaving well. For specific behaviours or tasks, this could also consist of fine-tuning on demonstrations of the desired behaviour by the game designer. The training procedure is equivalent to step 1.

3. **Generate Preference Data on Online Trajectories**
   Fine-tuned LLMs are subsequently prompted to generate multiple responses which are compared by human labellers to provide preferences. In the context of agents, the prompt becomes the initial observation (and optionally context of previous observations and actions provided). The agent is then rolled out from a given initial start state multiple times to collect multiple trajectories. Similarly to LLMs, the temperature of the softmax sampling of the policy for action selection can be increased to generate more diverse behaviours from the agent for easier comparison. A human (e.g. game designer) then provides preferences on these trajectories, such as preferring trajectories where the agent plays in a certain style.

4. **Train a Reward Model on Preferences**
   A reward model is then trained on these online trajectories such that the reward model provides higher reward for preferred trajectories, commonly using a Bradley Terry model (Bradley & Terry, 1952) for pairwise comparisons. While this reward model is usually trained from scratch in the context of agents, following the modern LLM procedure it is also possible to utilise the pre-trained or fine-tuned policy model by replacing the action classification head with a scalar regression head such that the reward model can also benefit from the pre-training, and share the same knowledge as the agent to reduce hallucinations (i.e. reduce out-of-distribution behaviours) (Ziegler et al., 2020).

5. **Align the Fine-tuned Model with the Reward Model**
   Finally the agent can be trained with online reinforcement learning, and optionally preference filtered fine-tuning, to maximise the reward provided by the reward model, thereby aligning the agent with the game designer's preferences. Since online reinforcement learning can be inefficient, further fine-tuning can be performed on the high reward trajectories to get the agent closer to the desired behaviour before deploying it online. A common failure mode at this stage is reward model over-optimisation, where the agent performs behaviours that maximise the reward model output but are not aligned with preferences (also known as reward hacking). If this occurs, regularisation towards the original policy can be added or steps 3-5 can be repeated to generate new preferences on the reward hacking behaviour which can be used to re-train the reward model. This may take multiple iterations (e.g. 5 reward model iterations were used for Llama2 (Touvron et al., 2023)), but eventually results in an aligned agent.

This procedure combines the benefits of large scale pre-training to obtain an informed and generalisable agent, with the benefits of reinforcement learning from preferences to obtain an agent reliably aligned with behaviour preferences. The online fine-tuning also helps to alleviate the well-known problems associated with imitation learning/offline agents going out of distribution (Ross et al., 2011) by refining the behaviour online.

# B  ARCHITECTURES AND TRAINING DETAILS

## B.1  BASE MODEL

For our base model ($\sim 103$M parameters) we use a GPT-2 causal transformer architecture with $8$ layers with $1024$ hidden dim. Each attention layer has $8$ heads, and the feedforward layers have a hidden dim of $4096$.

Each image is resized to be of shape $128 \times 128 \times 3$, divided by $255$ to put its values in $[0, 1]$, and is then fed into a convolutional encoder to map it to a $1024$ dimensional vector.

The first layer of the conv net has kernels of shape $8 \times 8$, with a stride of $4$, and a padding of $3$ and maps to $16$ channel dimension. This is followed by $4$ lots of ConvNext (Liu et al., 2022) and downsampling blocks (kernel of shape $3 \times 3$, stride of $2$, padding of $1$, doubling the channel dimension). Finally, a LayerNorm (Ba et al., 2016) is applied to the output.

The transformer operates on sequences of $32$ timesteps using learnt positional encodings.

The output of the transformer is layernormed, and then fed into an MLP with a single hidden layer of $1024$ dimensions with a GELU (Hendrycks & Gimpel, 2016) non-linearity.

For our optimiser we use AdamW (Loshchilov & Hutter, 2017) with a learning rate of $1e - 4$ and a weight decay of $1e - 4$. We use a batch size of $256$ with a learning rate warmup period of $1000$ updates and a gradient clipping value of $1$.

We train with the same image augmentations as used by (Baker et al., 2022), and filter out all no-op actions.

## B.2  FINE-TUNING OF BASE MODEL

We train for $1500$ batches of size $128$ with a learning rate of $1e - 6$ with the same image augmentations and no-op filtering as for pre-training with $200$ warmup steps.

## B.3  REWARD MODELS

Each trajectory is padded up to the maximum length of $100$ (with $0$ images) before being fed into the reward models.

For the `Random Encoder` model we randomly initialise a linear layer to randomly project the flattened values of the image to a $512$ dimensional vector. This linear layer is not trained.

For the `Agent Encoder` model, we feed the trajectory into the fine-tuned agent and take the layernormed output of the transformer, corresponding to timesteps $0$ up to $100$, as the $1024$ dimensional embeddings. The parameters of the fine-tuned agent are not trained.

For both models we then feed these vectors into an MLP with a GELU non-linearity and a hidden layer of $256$ and and output dimension of $3$. Each of the $100$ $3$-dimensional vectors are concatenated together and then fed into another MLP with a GELU, $256$ hidden dimension, and an output of $1$.

To train the reward models we use a minibatch of size $2048$, learning rate of $1e - 4$, and an $L_2$ regularisation penalty of $0.1$. We train all models for $200$ epochs, except for the largest training set size of $1000$ trajectories which we train for $50$ epochs.

After training, we compute the minimum and maximum outputs of the reward model on the training set. These are then used to normalise the output of the reward model to lie within $[0, 1]$.

### B.4  ALIGNMENT TRAINING

**Preference Fine-Tuning:** after training the reward model on its dataset of $M$ trajectories (which result in a dataset of up to $N = (M)(M-1)/2$ comparisions), we compute the reward for each of these $M$ trajectories. We then sort them by magnitude, and take the top $20\%$ of these as a smaller dataset to perform behaviour cloning on.

For this final step of BC, we use a learning rate of $1e-5$ with $1000$ updates on minibatches of size $256$. We only train the parameters of the MLP after the transformer layers.

**Reinforcement Learning:** in our experiments we use an undiscounted REINFORCE loss on batches of 16 episodes of up to 100 timesteps. We use a learning rate of $1e-4$ and once again only train the parameters of the MLP after the transformer layers. If an error occurs during an episode's rollout, we simply drop the samples from that episode and subsequently use a smaller batch size for the update.

## C  BLEEDING EDGE GAME HUMAN DATA COLLECTION

Human gameplay data was recorded as part of the regular gameplay process, in order to enable in-game functionality as well as to support future research. In game, recordings allowed players to view their past games to improve their skills and for entertainment. Games were recorded on the servers that hosted the games in the form of so-called replay files. Recordings include a representation of the internal game state and controller actions of all players.

## D  ABLATION OF UNSUPERVISED PRE-TRAINING AND MODEL SCALING

### D.1  ABLATION OF UNSUPERVISED PRE-TRAINING

We see from the results in Sections 4.1 and 4.2 that fine-tuning improves the task-specific performance of our pre-trained agent. However, to determine whether fine-tuning our base model is beneficial over simply training a model from scratch on our curated task-specific dataset, we also ablate the unsupervised pre-training stage. We train our agent from scratch for 20k updates, using the same procedure as used previously for fine-tuning (see Appendix B) until the loss appeared to converge. We subsequently evaluated the agent in the environment for 1000 episodes following the same procedure described in Section 4. The distribution of jumppads reached by the agent trained from scratch on the fine-tuning dataset is shown below (left) for comparison with the original pre-trained+fine-tuned agent (right) in Figure 9.

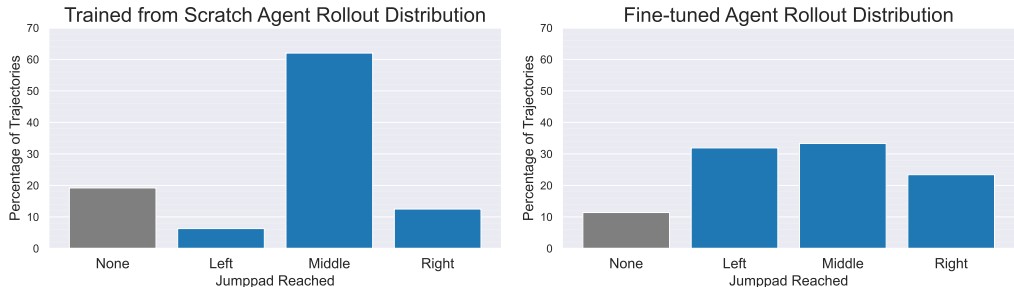

Figure 9: Distribution of jumppads reached by an imitation learning agent trained from scratch on the task-specific dataset used for fine-tuning the base model (left) compared to the base agent that was fine-tuned on this dataset.

We find that this agent has a greater failure rate (around 20% for the fine-tuned only agent fail to reach any jumppad compared to only around 10% for the pre-trained+fine-tuned agent) and a much narrower distribution of jumppads reached. This is surprising given the task-specific dataset consists of successful trajectories evenly split between the jumppads.

As additional anecdotal evidence for the benefit of unsupervised pre-training, we also noticed a fine-tuned agent miss a jumppad, hit the wall behind and then turn around to hit the jumppad. Since the

fine-tuning dataset consists of only curated trajectories that directly hit the dataset, this behaviour was not present in the fine-tuning dataset, and agents trained from scratch on the fine-tuned dataset are found to continuously run into the wall if they miss the jumppad, as they have never seen that observation before. This behaviour is shown in the supplementary videos of our agent at:

```
https://anonymous.4open.science/r/aligning-agents-like-llms/
Fine-Tuned%20Model/Fine-Tuned%20Missing%20but%20Turning%20Around.
mp4.
```

This provides further anecdotal but intuitive evidence for the benefits of unsupervised pre-training.

### D.2 PRELIMINARY MODEL SCALING ANALYSIS

To further justify the model size and investigate scaling properties of our transformer policy, we also trained smaller models of 4M and 25M parameters with equivalent architectures (described below in D.3) on our task-specific dataset, using the same procedure as above. The jumppad distributions for these models are provided in Figure 10.

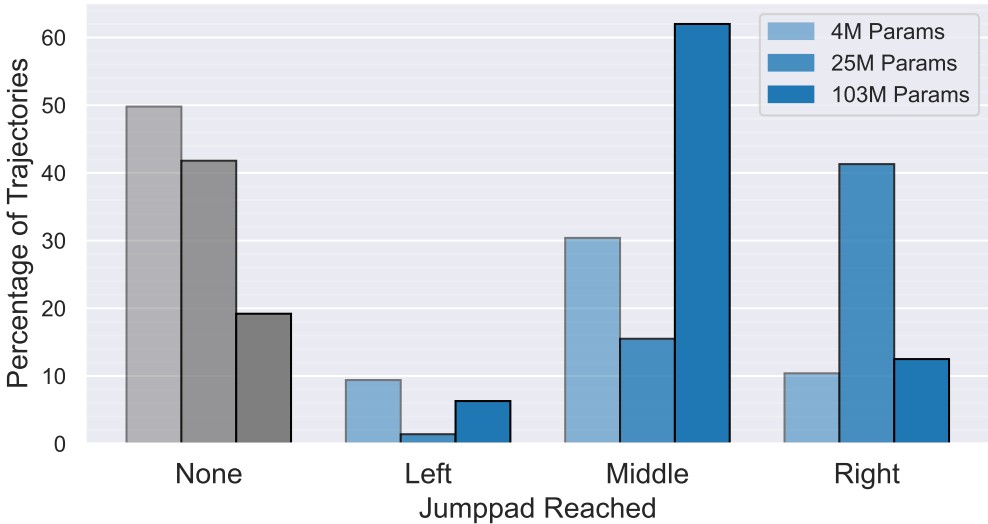

Figure 10: Distribution of jumppads reached by various size models (4M, 25M and 103M parameters) trained from scratch on the task-specific dataset used for fine-tuning the base model. Importantly, we see that task failure rates (the grey bars on the left) decrease as the number of parameters increases, even given the same size of dataset and number of training updates.

We see that these smaller models have much greater failure rates that the larger 103M parameter models shown in Figure 9, demonstrating that larger models are beneficial for imitation learning from pixels even on this relatively small task-specific dataset of 300 trajectories. While larger models still may be beneficial (particularly with pre-training on our large unsupervised dataset described in Section 4), larger models would further increase the crucial inference cost and we find that 103M parameters is sufficient for further alignment, as demonstrated in Figure 9.

### D.3 SMALL MODELS ARCHITECTURE

The architecture of our 2 smaller models is identical to that of the base model described in Section B, except for:

- ∼**4M:** 4 layers with 256 hidden dims and 4 heads for each attention layer.
- ∼**25M:** 8 layers with 512 hidden dims and 8 heads for each attention layer.
- (∼**103M:** 8 layers with 1024 hidden dims and 8 heads for each attention layer.)

### D.4 ALIGNMENT OF MODELS TRAINED FROM SCRATCH BY SIZE

To complete our ablation, we now investigate how pre-training and model size affects online alignment. To do so we followed our procedure in Appendix A and rolled these models out and generated preferences as in Section 4. We then trained the corresponding reward models and compare aligning these models trained from scratch to our pre-trained + fine-tuned model, as shown below in Figure 11.

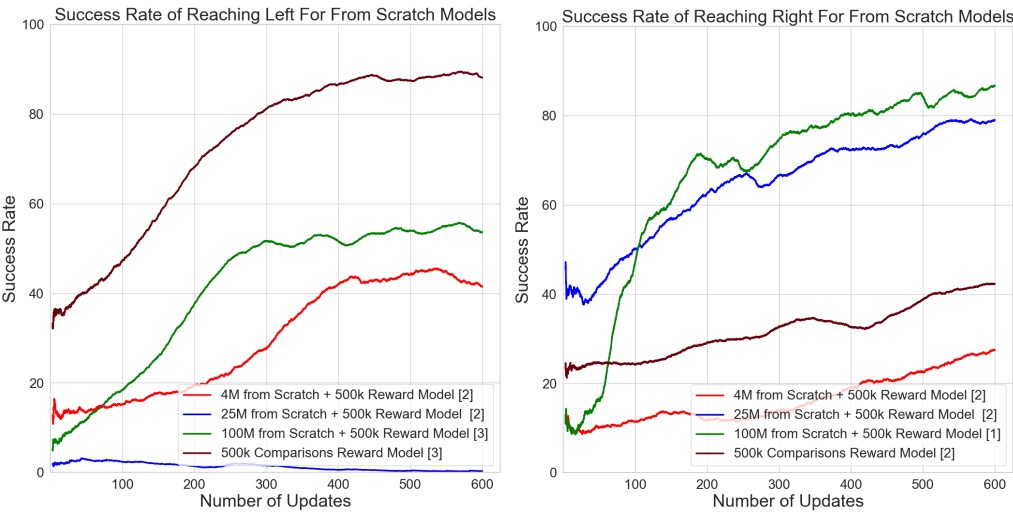

Figure 11: Left and right jumppad success rate during online alignment for models of different sizes trained from scratch using corresponding reward models trained on 500k comparisons.

We see that our pre-trained model aligns significantly better than the equivalent size model trained from scratch when aligned left, but worse when aligned right.

We also see that larger models generally align better in terms of increase in success rate during alignment, although the bias of the 25M parameter model towards going right make this trend less clear.

We note that in this experiment we only considered using the reward models trained on 500k comparisons, in order to see whether it was possible to successfully align a model trained solely on our small task-specific dataset. As noted in the previous section, these models have significantly less diversity in their behaviour than the model first pre-trained and then fine-tuned. This makes it much more costly (in terms of time to produce and subsequently provide labels) to generate sufficient examples of the desired behaviour (less than $10\%$ of trajectories successfully reached the left jumppad for the from scratch model for example compared to over $30\%$ for the pre-trained + fine-tuned agent).

## E    DISCUSSION OF OFFLINE TO ONLINE DISTRIBUTION SHIFT

As we mention in Section 4, the nature of using a real AAA video game is such that there is significant offline to online distribution shift between our offline training data and our online evaluation. One such source is due to character selection and customisable visual modifications, as shown below in Figure 12. Figures (a)-(d) are taken from our task-specific dataset and are representative of the input provided to our agent (only $256 \times 256$ rather than $128 \times 128$ resolution), while (e) shows the agent we use for evaluation, demonstrating the significant distribution shift in our environment.

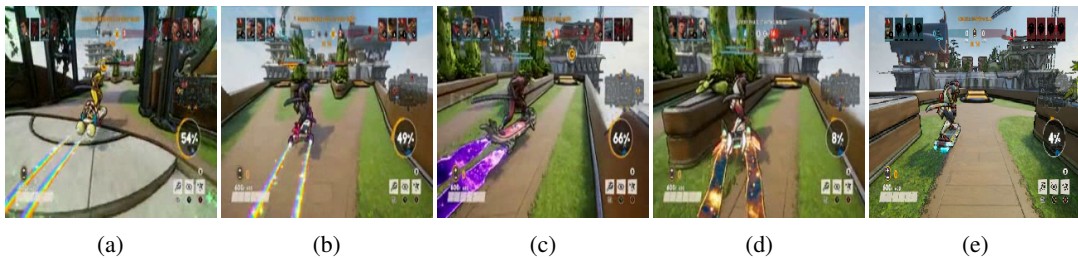

(a)        (b)        (c)        (d)        (e)

Figure 12: Screenshots of various agents with visual modifications performing our task contained within our general pre-training data. All are completely representative of the input provided to our agent (only $256 \times 256$ rather than $128 \times 128$ resolution). Figure (e) demonstrates the agent we use for online evaluation.

It is well-known that offline imitation learning often suffers from online distribution shift, particularly when learning from pixels in a partially observable environment, and due to action sampling (both of which we have in our setting) even without these significant visual changes, due to accumulating errors (Ross et al., 2011). The visual distribution shift in our environment only exacerbates these issues. As a result, imitation learning agents may not consistently perform imitated behaviours online even if the loss has converged on the offline data. This motivates the need to have some online fine-tuning (in our case from preferences) to refine the policy to consistently perform the desired behaviour online. Figure 4 demonstrates that even after the supervised fine-tuning stage on a dataset of trajectories that always reach a jumppad and involve the same character, a significant proportion of trajectories ($\sim 10\%$) do not reach any jumppad. However, after online fine-tuning, we are able to increase the success rate of the desired jumppad as shown in Figure 6, and the success rate overall for the left-aligned agent (see Appendix H).

Recent offline to online reinforcement learning literature has noted that there is often a performance drop when transferring an agent from the offline to online setting and provided approaches to mitigate this (Nair et al., 2021; Nakamoto et al., 2023; Ball et al., 2023). However, we highlight that these issues arise in offline RL due to the need to learn a pessimistic value function offline which is then not reflected by the true values found online. In our case we train only a policy offline (with imitation learning) and fine-tune this policy online (with on-policy REINFORCE from preferences). Therefore we do not utilise a value function, which means that we are able to achieve a smooth performance improvement online as demonstrated in Figure 6.

# F  REWARD MODEL PERFORMANCES BY JUMPPAD

Test reward model performances against number of comparisons used for training by jumppad are shown below in Figure 13.

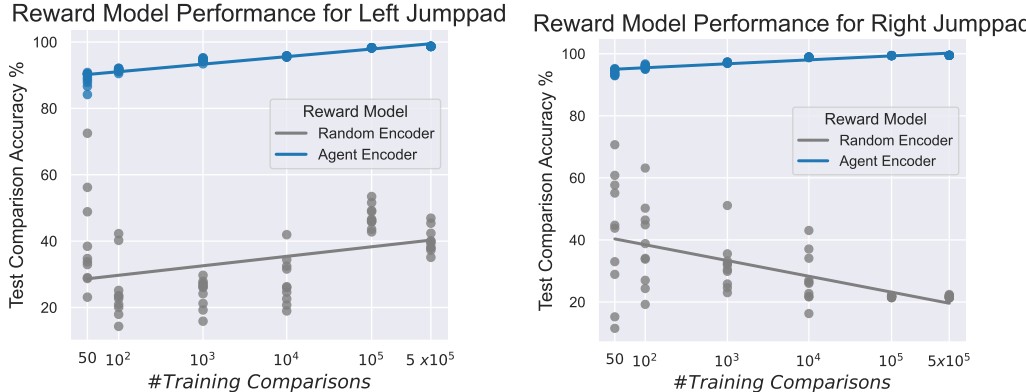

Figure 13: Test reward model performances against number of training comparisons by jumppad. Both jumppads show the same trend for the agent encoder, reaching $\sim 90\%$ performance for $100$ comparisons and $100\%$ performance by $500k$ comparisons. The random encoder is high variance, suggesting that the model does not have enough information to successfully generalise even for large numbers of training preferences.

# G   ALIGNMENT OF BASE MODEL FOR COMPARISON WITH FINE-TUNED MODEL

To understand how trajectory diversity affects alignment, we also consider aligning the base agent using the same reward models (trained on the fine-tuned agent). By plotting the successful trajectories that reach the left and right jumppad we see that the base agent (left) has a greater diversity than the fine-tuned agent (right). We note that from figures 3 and 4, the success rate for reaching the left and right jumppads is also lower.

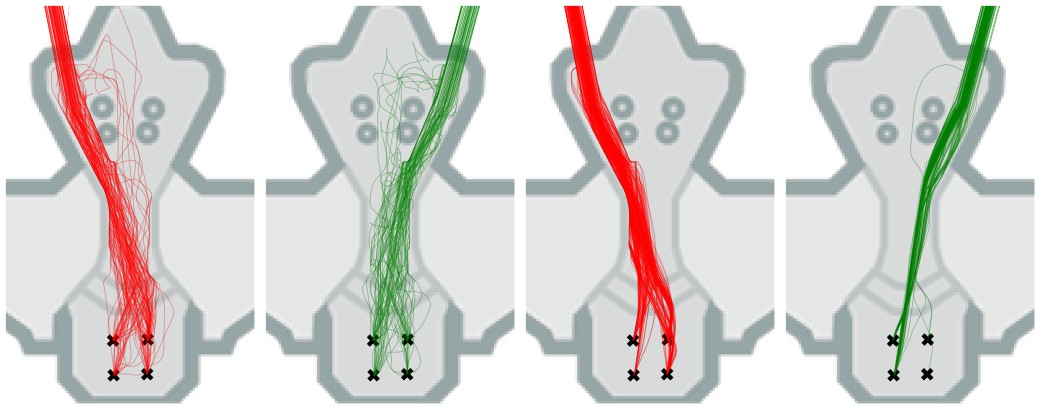

Figure 14: Base agent (left) and fine-tuned agent (right) trajectories for which the agents successfully reached the left (red) or right (green) jumppads. While fine-tuning provides a greater success rate, enabling more efficient preference labelling, it also reduces diversity of trajectories.

We now align the base agent to reach the left and right jumppads using the reward models trained on the fine-tuned agent, as shown below. We find that the agent can be aligned more symmetrically with both the left and the right jumppads, although alignment with the left jumppad still has a slightly better performance. However, in comparison to alignment of the fine-tuned agent (Figures 6 and 7), we see that the alignment is much slower, starting at a lower success rate, not increasing as quickly during training, and reaching a lower final success rate. As before, we see the same general trend that higher accuracy reward models (using more preference data) lead to better alignment.

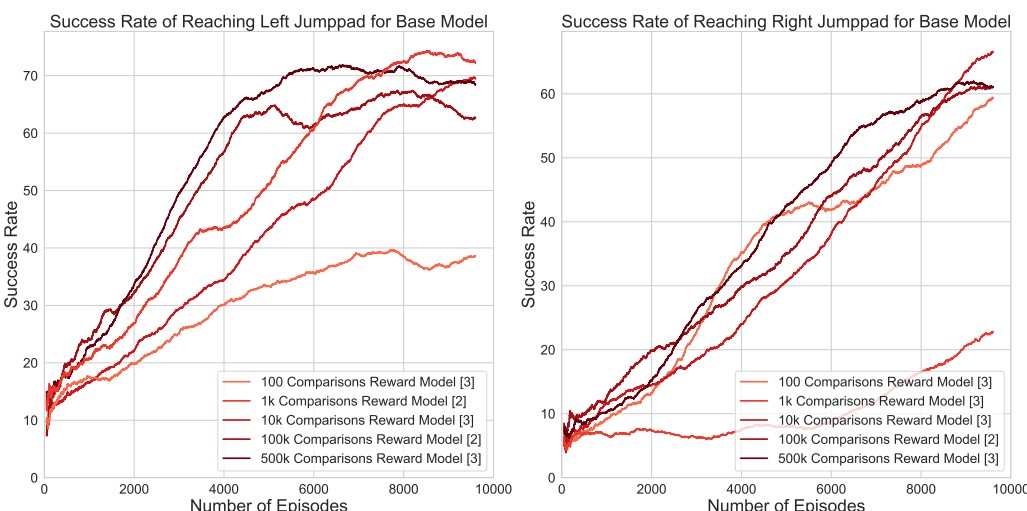

Figure 15: Left jumppad success rate for base agent using fine-tuned agent reward models.

Figure 16: Right jumppad success rate for base agent using fine-tuned agent reward models

# H    FINAL ALIGNED AGENT JUMPPAD DISTRIBUTIONS

Jumppad distributions for our final agents aligned to go left and right using preference fine-tuning and online alignment with reward models trained on 500k preferences are shown below.

First we partially align our agents with preference fine-tuning using our (500k comparison) reward models, so that the behaviour distributions are closer to the desired behaviour distributions to reduce the alignment required online, as shown in Figure 17.

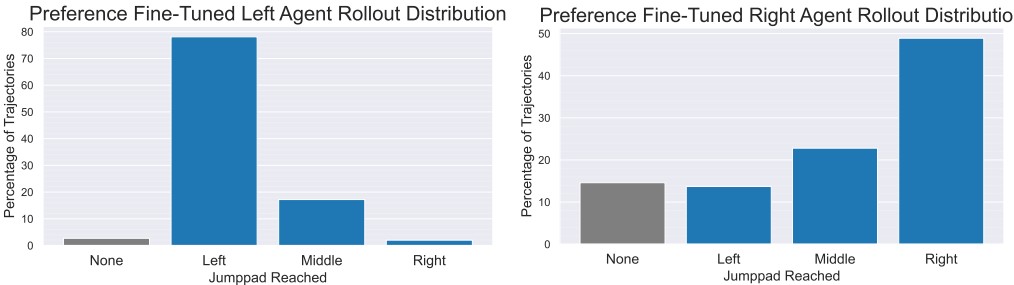

Figure 17: Left and right jumppad success rate for agents partially aligned with preference fine-tuning with reward models trained on 500k preferences.

We see that preference fine-tuning starts to align our agents towards the desired behaviour, but does not fully align our agents. Therefore we then perform online reinforcement learning using our (500k comparison) reward models until our agents are fully aligned.

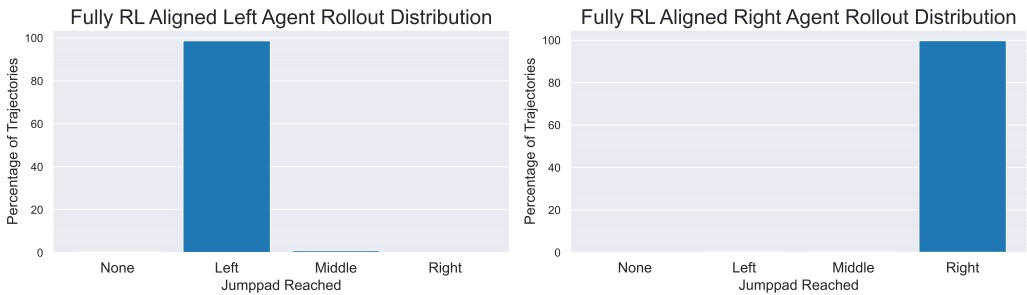

Figure 18: Left and right jumppad success rate for our fully aligned fine-tuned agents using preference fine-tuning and online reinforcement learning with reward models trained on 500k preferences.

Final evaluation shows that our agents have now been effectively fully aligned with our desired behaviour.

# I  HEATMAP AND VIDEOS OF GRADUAL ALIGNMENT OF AGENTS

To help visualise the gradual alignment of our agent, we provide a heatmap of the agent trajectories at each stage of our alignment pipeline below in Figure 19.

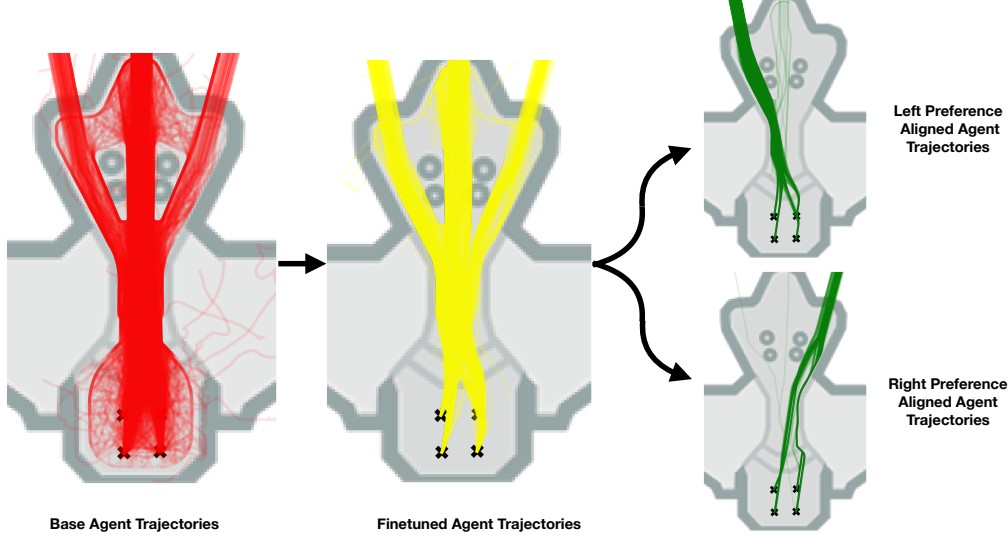

Figure 19: Heatmap of agent trajectories at each stage of our alignment pipeline. 1000 rollouts shown in each figure.

We see that the base agent trajectories are relatively diverse, capturing a variety of human behaviours, including those that do not reach a jumppad, or take a very indirect route to do so. The fine-tuned agent which has been refined on curated task-specific/demonstration trajectories shows more direct trajectories to the jumppads, but still does not incorporate any preference regarding the the jumppad which we would like our agent to reach. Finally, by training a reward model on these fine-tuned agent trajectories to capture our preferences, we are able to use that reward model for preference fine-tuning and subsequent online reinforcement learning to align our agent to reliably perform the desired behaviour, be it to reach the left or the right jumppad.

Videos of the behaviour of our agent at each stage of this alignment pipeline are provided at: `https://anonymous.4open.science/r/aligning-agents-like-llms`.

## J LIMITATIONS AND FURTHER WORK

This work provides a proof of concept for training agents to act as desired on a complex 3D environment. For our procedure to be applied in practice by game designers or in other domains such as robotics, the efficiency of this pipeline will be essential.

One limitation of our work is that we assume access to a large amount of human data on the game of interest to pre-train a base model. In practice however, when designing a game, large amounts of human data may not yet be available, and even for existing games it may not be possible to access. We leave investigation of the importance of the base model on downstream task performance, and the potential transfer of models pre-trained on other environments to future work.

Another limitation of our approach is that we utilise synthetic preferences. While this may be a realistic option for many use cases, human feedback from the game designer will be required in general. Given the time expense of providing feedback, especially for agents on modern console games which must often be run in real-time in order to be rendered, this process could be costly. Therefore supervised fine-tuning on relevant behaviours and efficient preference labelling will be essential. However, knowledge transfer from training LLMs could again be relevant here. Recent work on providing human feedback for LLMs has used a hybrid form of feedback that combines preference and evaluative feedback, in which responses are grouped into a batch of size $N$ and simultaneously compared on a preference scale of size $P$, which enables larger numbers of comparisons to be extracted from a given number of responses, improving the time efficiency of providing feedback. For example, InstructGPT (Ouyang et al., 2022) uses $4 \leq N \leq 9$, $P = 7$ while Llama2 (Touvron et al., 2023) uses $N = 2$, $P = 4$. As $N$ and $P$ increase, more information can be extracted from the provided human feedback. For example, with $N = 5$, $P = 5$ and assuming no category duplication (no trajectories are considered equal), $\binom{5}{2} = 10$ comparisons can be extracted, which is equivalent to 2 bits of information per trajectory watched, compared to just 0.5 bits per trajectory for default pairwise comparisons with $N = 2$, $P = 2$. This results in a $4\times$ improvement in feedback efficiency for human labellers, at the cost of potential label noise due to the additional mental overhead required. Similar strategies could be applied to providing feedback to agents to make the process more time efficient.

A final limitation is that running agents online in modern console games can be challenging. While our work demonstrated surprising efficiency for fine-tuning pre-trained agents with RL, for more complex behaviours and tasks this may become prohibitively expensive. Fortunately, developments in RLHF for language models can also be utilised here. We demonstrated in Section 4 that an intial step of preference fine-tuning could improve the efficiency of alignment, and explained that this corresponds to Reinforced Self-Training (ReST) (Gulcehre et al., 2023) with a single iteration. However, the full procedure using multiple iterations could be used to further improve the sample efficiency of aligning with preferences while maintaining the benefits of online exploration. Other recent work has investigated aligning models completely offline. Direct Preference Optimisation (Rafailov et al., 2023) provides an approach for optimising a model to align with preferences without the need for a reward model or online training, while (Hu et al., 2023) uses offline reinforcement learning with pre-generated samples and rewards. Additionally, efficient fine-tuning strategies such as LoRA (Hu et al., 2021) could be used to reduce the hardware requirements for the fine-tuning of a large base model for the individual applications of game designers. These new approaches provide promising directions for further work to improve the efficiency or even removing the need for online training completely.

