# OpenReview forum: "Aligning Agents like Large Language Models"
_ICLR.cc/2024/Conference — Submitted to ICLR 2024_

### Official Review · Reviewer_B6a1 · 2023-10-26

**Soundness:** 2 fair
**Presentation:** 3 good
**Contribution:** 2 fair
**Rating:** 5
**Confidence:** 4

**Summary:**

This paper investigates how to align large-scale imitation learning agents with supervised finetuning (SFT) and reinforcement learning from preferences. Specifically, this paper utilizes imitation learning for pretraining, supervised learning for finetuning, and REINFORCE for learning from preferences. Initial empirical findings show the effectiveness of the procedure of LLM for aligning agents in a complex 3D environment.

**Strengths:**

- A descent paper, easy to follow.
- Interesting idea to explore LLM technique for RL agent.

**Weaknesses:**

- Although the idea is interesting, the technique is too incremental, integrating IL for pretraining, IL for SFT, and REINFORCE for alignment.
- With the incremental technique, I would like to see more sufficient and interesting empirical findings. However, this paper does not contain the discussion including but not limited to:
1. Scaling Law. Whether model size scales, data scales, then the performance improves still holds for this setting?
2. In-context Learning. Whether preferences with some demonstrations in the model context can be captured?
3. Longer model context. Currently, the context length is 32 timesteps. Compared to typical LLMs with at least 2k steps, isn’t it too small to memorize useful information in the context?
4. During alignment, try PPO algorithm, which is verified as effective by many papers. Furthermore, why this paper utilizes an **undiscounted** version of REINFORCE? How about direct policy optimization (DPO) that utilizes supervised learning style alignment? As this paper only finetunes the last layer, how about full-parameter finetuning? How about LoRA?
5. Comparison and discussion of offline-to-online reinforcement learning.
6. Generalization. How about more environments or multi-task setting?

**Questions:**

Why is the imitation learning in the first stage claimed as "unsupervised pretraining" in Figure 1?

---

> ### Author Response · Authors · 2023-11-22
> **Author Response to Reviewer B6a1 (Part 1/2)**
>
> We thank you for your review and feedback on our work, and we’re pleased that you found the core idea interesting.
>
> > Although the idea is interesting, the technique is too incremental, integrating IL for pretraining, IL for SFT, and REINFORCE for alignment.
>
> Regarding novelty, we agree that the individual components of our work, and even the combination of imitation learning and reinforcement learning from preferences, is not novel, as discussed extensively in our related work in Section 2. However, there have since been many improvements to the basic pipeline of imitation learning followed by RLHF (introduced by [1]) in the context of large language models which have not been investigated back in the context of agents. Our main contribution is to apply this modern pipeline to a visually rich and complex domain with real human data, with the application of recent advances to improve the efficiency of the costly online alignment. For example:
>
>
> **1. Separation of imitation learning into pre-training on general data and fine-tuning on limited task-specific data**
>
> Pre-training on general text followed by instruction fine-tuning is essential for training helpful LLMs, but in the context of agents the usual procedure is to only imitate task-specific data. We demonstrate in Sections 4.1 and 4.2 that similar pre-training followed by task-specific fine-tuning is beneficial for task performance in our context for agents. We’ve also now added an additional ablation in Appendix D that demonstrates that the pre-training stage is beneficial over imitating the task-specific behaviour alone.
>
> **2. Use of Agent Model / Encoder for Initialising Reward Model**
>
> We demonstrate in Section 4.4 that utilising the agent model trained with an imitation learning loss as the initialisation for the reward model significantly improves the performance of the reward model and improves the feasibility of training a reward model with limited costly human feedback. While this is now typical for LLM reward models, the demonstration that this also applies in our case of reward modelling from pixels for decision-making agents across data sizes is a valuable contribution.
>
> **3. Introduction of Preference Fine-Tuning on Highest Reward Trajectories**
>
> We demonstrate in Section 4.5 that additional fine-tuning on a subset of the trajectories used for training the reward model with greatest reward significantly improves the efficiency of the subsequent alignment of the model with online reinforcement learning. This approach was concurrently introduced for language models [2], but is novel in the context of agents as far as we are aware. This is a big quality of life improvement for an end-user or game-designer that would like to align a (large) generalist agent, since fine-tuning on existing data is much less expensive than online reinforcement learning in a complex domain.
>
>
> Regarding other potential directions for investigation that you mention:
>
> > Scaling Law. Whether model size scales, data scales, then the performance improves still holds for this setting?
>
> Regarding the scaling of the model, we’ve also added additional results on how model size affects performance in our added ablation of the pre-training stage in Appendix D. As well as demonstrating the benefit of the unsupervised pre-training stage with supervised fine-tuning rather than training from scratch on the fine-tuning dataset, we found that larger model sizes (up to the 100M parameters we considered) are beneficial for our task even for training only on the more limited fine-tuning data.
>
>
> > In-context Learning. Whether preferences with some demonstrations in the model context can be captured?
> Longer model context. Currently, the context length is 32 timesteps. Compared to typical LLMs with at least 2k steps, isn’t it too small to memorize useful information in the context?
>
> Our context corresponds to 32 previous timesteps as you mention. Since this is sufficient to determine the agent’s current location and motion, it is possible for the agent to learn our desired navigation behaviour with this context. While longer context could be interesting in more general settings given the partial observability of the environment, we note that this would also be significantly more expensive in terms of compute and memory for little gain, given that there is a lot more information contained within each image of context than there is in each word of context.
>
> Investigating whether demonstrations could be incorporated in the agent’s context for in-context learning is an interesting idea, but out of scope of this work on efficiently aligning agents with preferences.

---

> > ### Author Response · Authors · 2023-11-22
> > **Author Response to Reviewer B6a1 (Part 2/2)**
> >
> > > During alignment, try PPO algorithm, which is verified as effective by many papers. Furthermore, why this paper utilizes an undiscounted version of REINFORCE? How about direct policy optimization (DPO) that utilizes supervised learning style alignment? As this paper only finetunes the last layer, how about full-parameter finetuning? How about LoRA?
> >
> > We utilised REINFORCE for simplicity and found it to be sufficient given we are only aiming to refine the policy to learn specific behaviours that are not too far from the distribution of the base model. We found discounting makes little difference as we want to reinforce entire preferred trajectories. In general however, using discounting and PPO to better control instability from large updates to the policy may be beneficial as we note in Section 4.5.
> >
> > Given our results on the effectiveness of preference fine-tuning, we hypothesise that Direct Policy Optimisation (DPO) could be very effective in our setting, but is out of scope of what we could properly analyse in a single conference paper and left for further work as discussed in Appendix J. Due to the offline to online distribution shift in our setting we expect that some online component is likely to be beneficial however, as discussed in the new Appendix E. Similarly we also believe LoRA could be effective in our setting as we mention in Section 4.5, but is also out of scope and left for further work as discussed in Appendix J. Full-finetuning was not possible due to the memory requirements of the dedicated hardware required to run the game online.
> >
> > > Comparison and discussion of offline-to-online reinforcement learning.
> >
> > We agree this is a good point for analysis that we had missed and we thank you for highlighting this. We have added a discussion of this in the new Appendix E (referenced from Section 4.1). We hope this explains the challenges of the significant offline to online distribution shift we have in our environment, and how our approach mitigates many issues associated with these challenges.
> >
> >
> > > Generalization. How about more environments or multi-task setting?
> >
> > Transfer to other environments is out of scope of this work, as discussed in Section 4.1 and Appendix J. Our work is multi-task in the sense that we aim for the base agent to capture diverse behaviour corresponding to many different tasks (such as navigating to different jumppads) and then aligning the agent to perform a particular task.
> >
> > > Why is the imitation learning in the first stage claimed as “unsupervised pretraining” in Figure 1?
> >
> > We follow the terminology used by the LLM community which follows from the original GPT paper [3]. In this paper, the "unsupervised pre-training" refers to next token prediction on a diverse "unsupervised" corpus of text data. As we explain in Appendix A, in our case we interpret this as next action prediction from our general game-play dataset where behaviours and goals are varied. "Supervised fine-tuning" then refers to the same next-token prediction objective but on "supervised" data, where the behaviour has been selected to provide the ‘correct’ or desired behaviour; in our case this corresponds to successfully navigating to a jumppad.
> >
> > We hope that the clarification of the novelty in our procedure and analysis, along with the additional ablation experiments we have provided, have helped to demonstrate the significance of our contribution towards bridging the gap between the LLM and gaming agent communities.
> >
> > **References:**
> >
> > [1] Reward learning from human preferences and demonstrations in Atari, https://arxiv.org/abs/1811.06521
> >
> > [2] Reinforced Self-Training (ReST) for Language Modeling, https://arxiv.org/abs/2308.08998
> >
> > [3] Improving Language Understanding by Generative Pre-Training, https://s3-us-west-2.amazonaws.com/openai-assets/research-covers/language-unsupervised/language_understanding_paper.pdf

---

> > > ### Comment · Reviewer_B6a1 · 2023-11-23
> > >
> > > Thanks for your feedbacks, though I disagree that some of the topics I mention are out of scope in this paper. I decide to keep my score unchanged.

---

### Official Review · Reviewer_Lh2V · 2023-10-27

**Soundness:** 1 poor
**Presentation:** 2 fair
**Contribution:** 1 poor
**Rating:** 3
**Confidence:** 3

**Summary:**

This paper proposes to apply the methodologies for alignment problem in large language models to game agents in a console game. The authors test the method in a console game (called Bleeding Edge) to align the behaviour of an agent follow human behaviour in a specific game mechanics to spawn off from one of the three starting points. The methodologies follows the InstructGPT procedures of 1. unsupervised pre-training the agent 2. Fine-tuning with demonstrated data 3. Using a reward function and align the deployed model for certain behaviour.

**Strengths:**

The paper is written nicely and easy to follow. The proposed game is interesting and different from the rest of the community.

**Weaknesses:**

No human behaviour data is going to be released. The game environment is not going to be released. This makes the reproducibility of the results impossible.

Limited details are provided of the environment, such as the internal mechanism of the game, how it simulates the physics. These critical details are essential given the game is not widely known in the research community.

The task is not particularly challenging and focus on a niche mechanics in the game. The task is only focusing on a small part of the full game, with a maximum rollout episode of 100 timesteps (equivalent of 10 seconds of game play). The task is comparably simpler than a lot of the locomotive control problems in the simulated experiments, with only 16-dim action space.

No benchmark is established for the task. Please compare the result with some basic model-free or model-based RL agents. Even classical control algorithms should be able to provide decent benchmarks.

Significant amount of work is needed for the paper to make the cut for the quality of ICLR.

**Questions:**

See weaknesses.

---

> ### Author Response · Authors · 2023-11-22
> **Author Response to Reviewer Lh2V (Part 1/2)**
>
> We thank you for your review and honest feedback on our work. We appreciate that you found the paper easy to follow, but would like to clarify our motivation and contribution to the community.
>
> > Weaknesses:
> >
> > No human behaviour data is going to be released. The game environment is not going to be released. This makes the reproducibility of the results impossible.
>
> Our main contribution is our analysis of applying the modern LLM training pipeline to such a challenging environment, which as you mention is interesting and different to those environments usually used by the community. On the other hand, as a result of being a real video game, there are many challenges and limitations of using Bleeding Edge as an environment, such as the hardware requirements (the game has to be run in real time on dedicated hardware for rendering) that make it unsuitable for release in its current state. As a result of these challenges, we do not claim to be providing a benchmark or dataset paper. However, we carefully provided our entire training procedure in Section 4 (and Appendix A) and all training details in Appendix B in order to make our results reproducible by practitioners looking to apply our procedure in other environments, even if it is not possible for them to be precisely replicated in Bleeding Edge.
>
> To be even more concrete, in addition to our general procedure the main contributions from our analysis are as follows:
>
> **1. Separation of imitation learning into pre-training on general data and fine-tuning on limited task-specific data**
>
> Pre-training on general text followed by instruction fine-tuning is essential for training helpful LLMs, but in the context of agents the usual procedure is to only imitate task-specific data. We demonstrate in Sections 4.1 and 4.2 that similar pre-training followed by task-specific fine-tuning is beneficial for task performance in our context for agents. We’ve also now added an additional ablation in Appendix D that demonstrates that the pre-training stage is beneficial over imitating the task-specific behaviour alone.
>
> **2. Use of Agent Model / Encoder for Initialising Reward Model**
>
> We demonstrate in Section 4.4 that utilising the agent model trained with an imitation learning loss as the initialisation for the reward model significantly improves the performance of the reward model and improves the feasibility of training a reward model with limited costly human feedback. While this is now typical for LLM reward models, the demonstration that this also applies in our case of reward modelling from pixels for decision-making agents across data sizes is a valuable contribution.
>
> **3. Introduction of Preference Fine-Tuning on Highest Reward Trajectories**
>
> We demonstrate in Section 4.5 that additional fine-tuning on a subset of the trajectories used for training the reward model with greatest reward significantly improves the efficiency of the subsequent alignment of the model with online reinforcement learning. This approach was concurrently introduced for language models [1], but is novel in the context of agents as far as we are aware. This is a big quality of life improvement for an end-user or game-designer that would like to align a (large) generalist agent, since fine-tuning on existing data is much less expensive than online reinforcement learning in a complex domain.
>
> > Limited details are provided of the environment, such as the internal mechanism of the game, how it simulates the physics. These critical details are essential given the game is not widely known in the research community.
>
> The environment we consider in this work is a real AAA video game, and so has relevant information publicly available [2,3] (including regarding the physics engine, PhysX [4]). We described what we believed to be the relevant aspects of the game for our research in Section 3, but have now also included links to these additional references in our paper, so thank you for highlighting that these additional details may also be of interest to the reader.
>
> **References:**
>
> [1] Reinforced Self-Training (ReST) for Language Modeling, https://arxiv.org/abs/2308.08998
>
> [2] https://www.bleedingedge.com/en
>
> [3] https://www.pcgamingwiki.com/wiki/Bleeding_Edge
>
> [4] https://www.nvidia.com/en-gb/geforce/technologies/physx/

---

> > ### Comment · Reviewer_Lh2V · 2023-11-23
> > **Maintaining the score**
> >
> > Thanks for the author's feedback.
> >
> > After reviewing the feedback and other reviews, I am maintaining my score. There exists major flaws in the set up of the problem given its limited experiments and diversity.

---

> ### Author Response · Authors · 2023-11-22
> **Author Response to Reviewer Lh2V (Part 2/2)**
>
> > The task is not particularly challenging and focus on a niche mechanics in the game. The task is only focusing on a small part of the full game, with a maximum rollout episode of 100 timesteps (equivalent of 10 seconds of game play). The task is comparably simpler than a lot of the locomotive control problems in the simulated experiments, with only 16-dim action space.
> >
> > No benchmark is established for the task. Please compare the result with some basic model-free or model-based RL agents. Even classical control algorithms should be able to provide decent benchmarks.
>
> Similarly, we feel that we should clarify the motivation of our work here: the navigation task we consider is representative of a behaviour that an end user might want an agent to complete. This task necessitates our proposed procedure since there is no reward function available. Therefore, how can we efficiently train an agent to complete such a behaviour? Even with access to the reward model to provide a single reward per trajectory (which already requires a reasonable policy/data to train), it would not be possible to use any model-free / model-based / classical control algorithms as baselines since they would never reach the goal to receive signal given we are training in an open 3D world from pixels only. Therefore we must use limited task-relevant / demonstration data for imitation learning. However, imitation learning may not provide exactly the preferred behaviour and is not very robust online due to distribution shift, particularly from pixels, necessitating final alignment with online preference-based reinforcement learning to reliably achieve the preferred behaviour, as demonstrated by our results in Section 4.
>
> We hope that our responses above have helped to clarify the contribution of our work, and that you can now better appreciate the value of our insights and contribution towards bridging the gap between the LLM and gaming agent communities.

---

### Official Review · Reviewer_S6ki · 2023-10-30

**Soundness:** 3 good
**Presentation:** 3 good
**Contribution:** 1 poor
**Rating:** 3
**Confidence:** 3

**Summary:**

Paper investigates the LLM-alike pretraining - instruction fine-tuning - RLHF scheme in game AI domains. Specifically, it focuses on a game called Bleeding Edge, and for each stage, the resulting agent is mainly evaluated on a target task where the goal is to reach left/middle/right jumping pad. The results show that 1) after instruction fine-tuning, the overall performance on the target task can be improved; 2) RLHF can help align the instruction-tuned model to a certain aspect (ex. left jumping pad) of the task but the aligned model can be hard to be re-aligned to some other aspects, echoing the importance of maintaining diversity during the alignment stage.

**Strengths:**

+The topic studied here is important. Albeit the success of the training-tuning-alignment scheme of large language models, its effectiveness in other domains has not been fully verified. This paper has made a valuable contribution to exploring this idea of agent learning in game AI and the results do show some promises. I believe it could drive the interest of audiences from both LLM and game AI communities (and possibly more).

+Although the task considered in this paper can be relatively simple (left/middle/right jumping pad), it is indeed investigated thoroughly, and yield some interesting (but not surprising) results, including the struggle on re-alignment.

**Weaknesses:**

Overall, I think the idea that this manuscript tries to put up with is clear and neat, but it can be a bit premature in terms of the width and depth of the investigation. Some substantial augmentation on the experiment part should be done before it can be accepted by a major conference. Here are some suggestions:

-Width: the authors have claimed they "investigate how the procedure for aligning LLMs can be applied to aligning agents from pixels in a complex 3D environment". Although I do agree that the environment can be visually perplexed, the task, however, might not be challenging enough. Is the goal hard to be recognized/identified due to the complextity of the 3D environment? Are there many distractions or interferences? Is the goal semantically rich therefore understanding it could become a challenge? Does the human preference over such goal go beyond simple multi-choice selection? Unfortunately, I find many of these aspects of interest when exploring pretraining - instruction-tuning - RLHF in game AI unchecked in this paper. Point is, this scheme has been verified in the language domain teaming with many of these aforementioned challenges and it actually work pretty well. Therefore, investigating this on a different domain (game AI), but without on par challenges, could be less illuminating to the community.

-Depth: although the paper offers rich variants of reward models in the RLHF experiments section, some aspects can still be missing. To name a few: the scale of the model(both the base model and the reward model), error range, etc. These will help with a better understanding on applying RLHF in a new domain.

**Questions:**

See "weaknesses"

---

> ### Author Response · Authors · 2023-11-22
> **Author Response to Reviewer S6ki (Part 1/2)**
>
> We thank you for your very constructive and helpful review. We’re pleased that you appreciate the importance of the topic, the valuable contribution we have made to exploring this idea, and our thorough investigation.
>
> To address your concerns regarding the scope of our analysis:
>
> > -Width: the authors have claimed they "investigate how the procedure for aligning LLMs can be applied to aligning agents from pixels in a complex 3D environment". Although I do agree that the environment can be visually perplexed, the task, however, might not be challenging enough. Is the goal hard to be recognized/identified due to the complextity of the 3D environment? Are there many distractions or interferences? Is the goal semantically rich therefore understanding it could become a challenge? Does the human preference over such goal go beyond simple multi-choice selection? Unfortunately, I find many of these aspects of interest when exploring pretraining - instruction-tuning - RLHF in game AI unchecked in this paper. Point is, this scheme has been verified in the language domain teaming with many of these aforementioned challenges and it actually work pretty well. Therefore, investigating this on a different domain (game AI), but without on par challenges, could be less illuminating to the community.
>
> We believe that our task is sufficiently challenging, and more importantly, illustrative for our analysis. Since we are using a real AAA video game with real human data from visual-only states, obtaining any behaviour reliably is challenging. The navigation task we considered is complex enough to necessitate our procedure, since there is no reward function to use for RL (and the preference reward model is too sparse for tabula rasa RL given the 3D environment), meaning RL alone is not possible. Therefore we must use limited task-relevant / demonstration data for imitation learning. However, imitation learning may not provide exactly the preferred behaviour and is not very robust to online distribution shift, particularly when learning from pixels, necessitating final alignment with online preference-based reinforcement learning to reliably achieve the preferred behaviour, as demonstrated by our results in Section 4. The task also still involves all the complexities of real human data, with the full visual input and gamepad action output spaces.
>
> As you highlight, this approach has demonstrated success in the language domain, but it is not trivial to apply it to agents in the game domain. In particular, there is significant offline to online distribution shift in our task due to character selection and visual modifications as we discuss in our new Appendix E, as well as the context (equivalent to prompting) that the agent is provided with at initialization as we discuss in Section 4.1. Additionally, learning from pixels and in an environment in which online reinforcement learning can only be performed in real time (due to game engine constraints) provides a much more challenging domain than language benchmarks. Our work therefore demonstrates what can be learned and applied from progress in the language domain, while highlighting additional challenges in the game domain, as in Section 4.5.

---

> > ### Author Response · Authors · 2023-11-22
> > **Author Response to Reviewer S6ki (Part 2/2)**
> >
> > > -Depth: although the paper offers rich variants of reward models in the RLHF experiments section, some aspects can still be missing. To name a few: the scale of the model(both the base model and the reward model), error range, etc. These will help with a better understanding on applying RLHF in a new domain.
> >
> >
> > Our main contribution is to apply the modern LLM training pipeline to a visually rich and complex domain with real human gameplay data, with the application of multiple recent advances this pipeline for language models back to agents to improve the efficiency of the costly online alignment. Some of the contributions of our analysis of applying RLHF in this domain included:
> >
> > **1. Separation of imitation learning into pre-training on general data and fine-tuning on limited task-specific data**
> >
> > Pre-training on general text followed by instruction fine-tuning is essential for training helpful LLMs, but in the context of agents the usual procedure is to only imitate task-specific data. We demonstrate in Sections 4.1 and 4.2 that similar pre-training followed by task-specific fine-tuning is beneficial for task performance in our context for agents. We’ve also now added an additional ablation in Appendix D that demonstrates that the pre-training stage is beneficial over imitating the task-specific behaviour alone.
> >
> > **2. Use of Agent Model / Encoder for Initialising Reward Model**
> >
> > We demonstrate in Section 4.4 that utilising the agent model trained with an imitation learning loss as the initialisation for the reward model significantly improves the performance of the reward model and improves the feasibility of training a reward model with limited costly human feedback. While this is now typical for LLM reward models, the demonstration that this also applies in our case of reward modelling from pixels for decision-making agents across data sizes is a valuable contribution.
> >
> > **3. Introduction of Preference Fine-Tuning on Highest Reward Trajectories**
> >
> > We demonstrate in Section 4.5 that additional fine-tuning on a subset of the trajectories used for training the reward model with greatest reward significantly improves the efficiency of the subsequent alignment of the model with online reinforcement learning. This approach was concurrently introduced for language models [1], but is novel in the context of agents as far as we are aware. This is a big quality of life improvement for an end-user or game-designer that would like to align a (large) generalist agent, since fine-tuning on existing data is much less expensive than online reinforcement learning in a complex domain.
> >
> > The combination of these additions in this domain makes it possible to align agents with only limited preference data and online RL updates.
> >
> > Regarding the scaling of the model, we’ve also added additional results on how model size affects performance in our added ablation of the pre-training stage in Appendix D. As well as demonstrating the benefit of the unsupervised pre-training stage over supervised fine-tuning only, we found that larger model sizes (up to the $\sim100M$ parameters we considered) are beneficial for our task even for training only on the more limited fine-tuning data.
> >
> > We hope that we have addressed your concerns and clarified the value of our contributions towards bridging the gap between the LLM and gaming agent communities.
> >
> >
> > **References:**
> >
> > [1] Reinforced Self-Training (ReST) for Language Modeling, https://arxiv.org/abs/2308.08998

---

### Official Review · Reviewer_SbPz · 2023-10-31

**Soundness:** 3 good
**Presentation:** 2 fair
**Contribution:** 2 fair
**Rating:** 5
**Confidence:** 4

**Summary:**

Similar to recent work in finetuning LLMs to align them with human preferences, the authors propose a process for aligning game agents that have been initially pretrained on a diverse, multimodal dataset via imitation learning. The paper studies the specific setting of taking an agent pretrained on diverse human behaviors in a 3D game (Bleeding Edge), and finetuning the agent policy to focus on only navigating to a chosen one of the three possible locations. The overall process consists of initial pretraining on a diverse dataset of human behaviors, next finetuning on a smaller set of curated, high quality demonstrations for a particular behavior, then finally finetuning the policy with a learned preference-based reward model.

**Strengths:**

- The overall motivation of studying how well imitation-learning + RL with preference-based reward models applies aligning generally capable game agents is interesting and meaningful.
- The additional experiments showing the efficacy of combining online learning with the learned reward model and additional finetuning on the top 20% of offline trajectories (as ranked by the same reward model) is interesting and a meaningful contribution, as most prior works only focus on either the offline or online settings individually rather than combining them.
- The experiments and discussion are thorough and detailed – in particular, it’s helpful to note why the policy finetuned for the right jumppad was less effective, despite the symmetry of the map, and to clearly show the gap in performance when building a reward model off of the pretrained policy encoder rather than using a random one.

**Weaknesses:**

- The paper shows that the same overall process for finetuning LLMs can also be applied to agents within this game. However, the overall method has limited novelty – as noted in the related works section, the general process of bootstrapping RL with imitation learning and further finetuning agent behaviors with reward models learned from human preferences have both been studied in prior work training RL agents outside of LLMs.
- The authors note that unlike prior work, their goal is to train "a generally capable base agent that can be aligned by game designers or end-users to perform different tasks or behave with different styles". This is a meaningful contribution even if the overall process itself is not novel, however, I believe that the current set of behaviors studied in this work (navigating to 3 different locations) is too limited. The paper would be greatly strengthened if the sets of behaviors or tasks demonstrated by the base and fine-tuned agents were more complex. While I understand the merit of first studying the approach in a simpler setting to disentangle what the agent is learning at each stage, the paper would be more complete with additional experiments showing the methodology also works on learning more complex behaviors, as this is the underlying motivation of the work.
- Rather than just having the bar plots in the first two stages and then reward curves for the final stage, the paper presentation would be strengthened with an overarching connecting figure showing a heat-map of how the agent behaviours across the map evolve at each stage (first showing diverse locations visited, then focusing on the jumppads evenly, then focusing on just the left/right one).

**Questions:**

While the more poor performance from the agent finetuning to the right jumppad is likely due to the lack of data and diversity as noted by the authors, this is also exactly the problem that RL should be able to address. Have the authors tried more sophisticated RL algorithms for encouraging diverse exploration (adding in an entropy bonus, injecting randomness, more complex intrinsic rewards on top of the human preferences)?

---

> ### Author Response · Authors · 2023-11-22
> **Author Response to Reviewer SbPz (Part 1/2)**
>
> We thank you for your comprehensive review and helpful suggestions. We’re pleased that you appreciate the meaningful motivation, the multiple contributions of our paper, and our thorough experiments and discussion.
>
> To address your concerns:
>
> > The paper shows that the same overall process for finetuning LLMs can also be applied to agents within this game. However, the overall method has limited novelty – as noted in the related works section, the general process of bootstrapping RL with imitation learning and further finetuning agent behaviors with reward models learned from human preferences have both been studied in prior work training RL agents outside of LLMs.
>
> Regarding novelty, we agree that the individual components of our work, and even the combination of imitation learning and reinforcement learning from preferences, is not novel, as discussed extensively in our related work in Section 2. However, there have since been many improvements to the basic pipeline of imitation learning followed by RLHF (introduced by [1]) in the context of large language models which have not been investigated back in the context of agents. Our main contribution is to apply this modern pipeline to a visually rich and complex domain with real human data, with the application of recent advances to improve the efficiency of the costly online alignment. For example:
>
> **1. Separation of imitation learning into pre-training on general data and fine-tuning on limited task-specific data**
>
> Pre-training on general text followed by instruction fine-tuning is essential for training helpful LLMs, but in the context of agents the usual procedure is to only imitate task-specific data. We demonstrate in Sections 4.1 and 4.2 that similar pre-training followed by task-specific fine-tuning is beneficial for task performance in our context for agents. We’ve also now added an additional ablation in Appendix D that demonstrates that the pre-training stage is beneficial over imitating the task-specific behaviour alone.
>
> **2. Use of Agent Model / Encoder for Initialising Reward Model**
>
> We demonstrate in Section 4.4 that utilising the agent model trained with an imitation learning loss as the initialisation for the reward model significantly improves the performance of the reward model and improves the feasibility of training a reward model from pixels with limited costly human feedback. While this is now typical for LLM reward models, the demonstration that this also applies in our case of reward modelling from pixels for decision-making agents across data sizes is a valuable contribution.
>
> **3. Introduction of Preference Fine-Tuning on Highest Reward Trajectories**
>
> We demonstrate in Section 4.5 that additional fine-tuning on a subset of the trajectories used for training the reward model with greatest reward significantly improves the efficiency of the subsequent alignment of the model with online reinforcement learning. This approach was concurrently introduced for language models [2], but is novel in the context of agents as far as we are aware. This is a big quality of life improvement for an end-user or game-designer that would like to align a (large) generalist agent, since fine-tuning on existing data is much less expensive than online reinforcement learning in a complex domain.
>
> We hope these three main examples highlight the novelty of our contribution in the context of aligning agents.
>
> **References:**
>
> [1] Reward learning from human preferences and demonstrations in Atari, https://arxiv.org/abs/1811.06521
>
> [2] Reinforced Self-Training (ReST) for Language Modeling, https://arxiv.org/abs/2308.08998

---

> > ### Author Response · Authors · 2023-11-22
> > **Author Response to Reviewer SbPz (Part 2/2)**
> >
> > > The authors note that unlike prior work, their goal is to train “a generally capable base agent that can be aligned by game designers or end-users to perform different tasks or behave with different styles”. This is a meaningful contribution even if the overall process itself is not novel, however, I believe that the current set of behaviors studied in this work (navigating to 3 different locations) is too limited. The paper would be greatly strengthened if the sets of behaviors or tasks demonstrated by the base and fine-tuned agents were more complex. While I understand the merit of first studying the approach in a simpler setting to disentangle what the agent is learning at each stage, the paper would be more complete with additional experiments showing the methodology also works on learning more complex behaviors, as this is the underlying motivation of the work.
> >
> > In order to provide a meaningful analysis of what is beneficial to transfer from current LLM training pipelines to align agents with user preferences, we intentionally utilise a minimal environment with a quantifiable metric to disentangle the effects of each stage of the pipeline. Our current task provides such an environment which maintains the complexities of the full game, with clear interpretation of how each stage of the training procedure affects the jumppad distribution. The task still *necessitates our procedure* however, since there is no reward function to use for RL (and the preference reward model is too sparse for tabula rasa RL given the 3D environment), meaning RL alone is not possible. Therefore we must use limited task-relevant / demonstration data for imitation learning. However, imitation learning may not provide exactly the preferred behaviour and is not very robust to online distribution shift, particularly when learning from pixels, necessitating final alignment with online preference-based reinforcement learning to reliably achieve the preferred behaviour, as demonstrated by our results in Section 4. The task also still involves all the complexities of real human data involving many characters and visual modifications and associated online distribution shift, with the full visual input and gamepad action output spaces.
> >
> >
> > > Rather than just having the bar plots in the first two stages and then reward curves for the final stage, the paper presentation would be strengthened with an overarching connecting figure showing a heat-map of how the agent behaviours across the map evolve at each stage (first showing diverse locations visited, then focusing on the jumppads evenly, then focusing on just the left/right one).
> >
> > We agree that a summary heatmap demonstrating how agent behaviours evolve during alignment is useful addition to demonstrate our core contribution, so thank you for this helpful suggestion. We have added this figure to Appendix H in our paper and referenced it from Section 4.5, with colours corresponding to our original Figure 1.
> >
> > To address your questions:
> >
> > > While the more poor performance from the agent finetuning to the right jumppad is likely due to the lack of data and diversity as noted by the authors, this is also exactly the problem that RL should be able to address. Have the authors tried more sophisticated RL algorithms for encouraging diverse exploration (adding in an entropy bonus, injecting randomness, more complex intrinsic rewards on top of the human preferences)?
> >
> > Since our environment is a real AAA video game involving an open 3D world, which must be run in real time for online RL, significant exploration would be infeasibly expensive (particularly for end users). Our motivation is instead to capture a diverse multi-modal distribution of human behaviour, before refining this behaviour through fine-tuning and aligning with preferences to select specific behaviours from this initial base distribution. This is what enables us to align our agent in only 600 RL updates. We believe that a more effective approach would be to have less aggressive fine-tuning, or to fine-tune on a larger task-relevant dataset to prevent this premature loss of diversity in our pipeline, as we discuss in Section 4.5.2. Further improving the sample efficiency of online alignment with additional offline training would be a good direction for further work.
> >
> > We hope this helps to explain the challenges of our environment and motivation for our approach, as well as the value of our contribution towards bridging the gap between the LLM and gaming agent communities.

---

### Author Response · Authors · 2023-11-22
**Author Summary of Improvements following Reviewer Feedback**

We sincerely thank all of the reviewers for taking the time to review our paper and to provide helpful feedback which we have now incorporated into our paper. The main additions to our work are as follows:

1. An ablation of the unsupervised pre-training stage, to demonstrate the benefit of pre-training on a larger environment dataset before fine-tuning on a task-specific dataset, provided in Appendix D.
2. A preliminary model scaling investigation, demonstrating that larger models (up to the 103M parameters that we use in the paper) perform better on our task, even when only trained on the comparatively small fine-tuning dataset. Also provided in Appendix D.
3. A discussion of the offline to online distribution shift challenges in our environment in Appendix E.
4. A heatmap of the agent distribution at each stage of the alignment pipeline in Appendix H.
5. Videos of our agents at each stage of the alignment pipeline at: https://anonymous.4open.science/r/aligning-agents-like-llms.

We have also provided responses to each reviewer below to address individual questions and concerns.

---

### Meta-Review · Area_Chair_8gw4 · 2023-12-11

**Metareview:**

The paper uses current paradigms of pretraining , instruction fine tuning and RLHF used in LLM alignment, to align AI agent in 3D gaming environment so that  the agent complies with a particular target behavior.

While reviewers agreed that the paper promise is interesting in terms of exploring these LLM alignment ideas  for agent alignment from pixels in the gaming domain, they found 1) the contribution of the work to be limited,  2) the experimentation to be narrow in terms of its scope  and in terms of the baselining of the methods.  Another pain point was that the paper is not easy to reproduce since data and code were not made available.

Authors provided during the rebuttal ablation study for showing benefits of pretraining the model, as well as initial scaling laws and added a discussion of the offline to online distribution shift and a heatmap suggest by reviewer SbPz.

As suggested by the reviewers expanding the scope of the paper to other gaming and more complex alignment behaviors and adding simple baselines to compare to  as well as comparing RLHF and DPO strategies of alignment will benefit the paper and make it more appealing and ready for publication.

**Justification For Why Not Higher Score:**

The methodological contribution of the paper is limited and the validation/ experimentation has a very narrow scope.

**Justification For Why Not Lower Score:**

N/A

---

### Decision · Program_Chairs · 2024-01-16

Reject